microbiology, molecular biology, genomics

microbiome, canine, industrialization, ancient DNA

**Authors for correspondence:**
K. Yarlagadda
e-mail: karthik9yar@gmail.com
R. S. Malhi
e-mail: malhi@illinois.edu

# Geographically diverse canid sampling provides novel insights into pre-industrial microbiomes

K. Yarlagadda[1], A. J. Zachwieja[6], A. de Flamingh[2], T. Phungviwatnikul[3], A. G. Rivera-Colón[4], C. Roseman[5], L. Shackelford[1], K. S. Swanson[3] and R. S. Malhi[1,2,4,5]

[1]Department of Anthropology, [2]Carl R. Woese Institute for Genomic Biology, [3]Department of Animal Sciences, [4]Department of Evolution, Ecology, and Behavior, and [5]School of Integrative Biology, University of Illinois Urbana–Champaign, Urbana, IL, USA
[6]Department of Biomedical Sciences, University of Minnesota Medical School Duluth, Duluth, Minnesota, USA

 KY, 0000-0003-2581-0782; AJZ, 0000-0003-3276-071X; AGR-C, 0000-0001-9097-3241; KSS, 0000-0001-5518-3076

Canine microbiome studies are often limited in the geographic and temporal scope of samples studied. This results in a paucity of data on the canine microbiome around the world, especially in contexts where dogs may not be pets or human associated. Here, we present the shotgun sequences of fecal microbiomes of pet dogs from South Africa, shelter and stray dogs from India, and stray village dogs in Laos. We additionally performed a dietary experiment with dogs housed in a veterinary medical school, attempting to replicate the diet of the sampled dogs from Laos. We analyse the taxonomic diversity in these populations and identify the underlying functional redundancy of these microbiomes. Our results show that diet alone is not sufficient to recapitulate the higher diversity seen in the microbiome of dogs from Laos. Comparisons to previous studies and ancient dog fecal microbiomes highlight the need for greater population diversity in studies of canine microbiomes, as modern analogues can provide better comparisons to ancient microbiomes. We identify trends in microbial diversity and industrialization in dogs that mirror results of human studies, suggesting future research can make use of these companion animals as substitutes for humans in studying the effects of industrialization on the microbiome.

## 1. Introduction

Industrialization represents a shift in both diet and environment, key factors in microbial variability that have come under increased scrutiny as microbiome studies have spread across more diverse populations. In humans, industrialization is associated with a reduction in microbiome diversity [1–12]. One of the challenges in investigating this reduction, however, is the sparse global sampling of microbiomes—there is a wealth of microbiome data from industrialized populations, but far fewer studies that have worked with the microbiomes of non-industrialized populations. Microbiomes are also complex products interacting within complex systems, and experimental studies in non-laboratory settings are challenging. Industrialization is a compound factor that is difficult to disentangle—how much do diet, environment or other factors associated with industrialized populations drive the observed differences? There are no simple solutions to these problems, but we can begin to chip away at them through increased global sampling efforts, international collaboration, and by taking advantage of existing models. Here, we present the analysis of a large set of

novel canine fecal microbiome data that characterizes shifts in microbiomes across global populations over centuries in time, and tests the question of diet versus environment in a controlled laboratory experiment.

## (a) Global canine microbiomes

The canine fecal microbiome has been the subject of a great deal of study, mostly centred around shifts in diet and resulting changes in fecal metabolites and microbes [13]. Dogs are ideal candidates for understanding shifts in the human microbiome in many respects: they are frequently companion animals and share human environments; their diet (in pet contexts) is consistent, the composition quantifiable by macronutrient, and they share an evolutionary history with humans, as through domestication, dogs underwent genetic, dietary and microbial shifts [14–25]. Recent canine microbiome studies have even recapitulated the results from human studies of microbiome responses to dietary shifts [26–29]. These aspects allow for controlled experiments and a more nuanced understanding of the canine fecal microbiome, how it responds to differences in diet, environment and disease, and even how these trends can be extended to other mammals—including humans.

Despite this potential, studies of dog fecal microbiomes have been relatively limited in geographical scope [30]. Much like studies of human genomes, limited diversity of samples hinders our understanding of the broader context within which results are placed and can lead to inaccurate inferences [31]. For example, numerous general microbiome studies support the positive relationship between the genus *Prevotella* and dietary fibre and the genus *Bacteroides* with dietary protein—but it is unknown if these trends hold in pet populations in non-industrialized contexts [21,27,32–40]. Similarly, the lack of data on strays or feral dogs hinders our understanding of how social interaction with humans and human-associated diets influence canine microbiomes. At best, we have a few studies that compare the microbiomes of dogs and wolves, but there are known genetic changes between wolves and dogs that would affect their diet, and thus microbiome [41]. While it may be safe to posit that, as for all studied mammals to date, the fecal microbiome plays some role in digestion and energy production for the host, the method by which this occurs (the functional part of the microbiome) and the bacteria responsible for these behaviours (the taxonomic part of the microbiome) remain unknown in most canine populations [13,21,42,43].

To better understand the impact of industrialization on canine microbiomes, we first need to sample more broadly across geographical populations. By establishing a rough landscape of what variation in the canine fecal microbiome looks like around the world, we can then begin to investigate the impact of specific alterations, like changes in diet and environment. A diverse sampling scheme also provides opportunity to explore the complex variation in industrialization around the world and better identify its influence on canine microbiomes, which are just as exposed to these shifts in diet and environment as their human counterparts. To achieve this, we sampled fecal microbiomes from dogs in South Africa, India and Laos, each representing a varying level of industrialization, capturing a more diverse representation of canine fecal microbiomes.

## (b) Canine microbiomes over time

Another area of microbiome research that has seen growth is the study of ancient microbiomes. Dental calculus and coprolites represent ancient, preserved biomaterials of both host genetic material and the microbes of those regions (the oral and fecal microbiome, respectively) [44,45]. While studies of coprolites have demonstrated a great deal of variability in genetic preservation of microbes, as well as challenges in the contamination of these samples, progress has been made in identifying portions of these ancient microbiomes [46–58]. However, as noted previously, these studies are once again hindered by the lack of diversity in modern microbiome studies [30,49]. With mostly modern, industrialized pets represented in studies, our definition of what constitutes a 'normal' canine microbiome is skewed, which limits our ability to successfully discriminate between ancient contaminants and ancient microbes of interest [59,60]. The dogs represented in these ancient studies are frequently human-associated, post-domestication; but their lifestyles, environment and diet mirror none of the dogs studied today [61–66]. Studies of dogs fed raw food diets are perhaps the best available facsimile for these ancient studies, but they still fail to account for the lack of industrialized environment [18,32,67–73].

We address this problem with our first step; by sampling globally diverse populations of dogs, some pets and some strays, we have a better representation of the canine fecal microbiome. This allows us to better analyse ancient canine fecal microbiomes, with the hypothesis that these ancient samples will be more similar to modern canine microbiomes from non-industrialized contexts. In fact, the dogs from Laos, who are human-associated but consume a mix of agricultural products and foraged foods, are likely to have the most similar fecal microbiomes due to their outdoor environment and mixed diet. Because industrialization would not have affected the ancient dogs, similarities between ancient and modern dog fecal microbiomes also provide insight into the question of industrialization's impact on the microbiome. Modern dogs in industrialized contexts are expected to have less diverse and less similar microbiomes to both non-industrialized modern populations' microbiomes and the non-industrialized ancient population's microbiome.

## (c) Diet versus environment

Working with dogs provides a unique opportunity to test the influence of diet and environment (local exposure to microbes from other animals, the natural and built world) on the canine fecal microbiome in a way that is difficult to replicate in the field. In our first step, we sampled canine fecal microbiomes from around the world in diverse contexts—where both diet and environment are different to what is abundant in the literature. To understand how these two factors can shape the microbiome, we can conduct a dietary experiment, using a well-established methodology that is frequently employed in canine microbiome studies. By sampling laboratory dogs in a frequently sterilized and indoor environment, we establish a baseline for the commonly sampled, industrialized canine microbiome. We can then feed these dogs a diet similar to one of our global canine populations—in this case, the diet of the dogs from Laos—replicating their diet, but not their environment. The resulting observed similarity between the microbiomes of the industrialized dogs on a diet similar to the dogs from Laos and either their baseline microbiomes or the microbiomes from the dogs from Laos indicates whether environment or diet plays a greater role in influencing the canine fecal

microbiome. This experiment adds a much-needed level of nuance to studies of microbiomes and industrialization.

In summary, we present a multi-population study of canine microbiomes from around the world, covering multiple contexts and environments with a shotgun sequencing approach, which produces both taxonomic and functional datasets. We look to examine this data within the framework laid out here, to better understand the impact of industrialization on the microbiome. First, we identify what diversity in the global canine microbiome population looks like, testing the existing hypothesis that increasing degrees of industrialization reduces microbiome diversity, but in dogs. We follow this with a comparison to ancient canine microbiomes, with the hypothesis that non-industrialized modern canine populations provide a better representation of the microbiome observed in the ancient dogs. Finally, we conduct a dietary experiment, feeding dogs in the United States (US) a diet similar to the dogs from Laos, to identify whether diet or environment has a greater influence on the fecal microbiome.

## 2. Methods

### (a) Global canine microbiomes

We selected three populations of dogs in variably industrialized contexts for this study from India, Laos and South Africa (electronic supplementary material, table S1). In all cases, fecal samples were collected in duplicate and were stored on FTA cards (Whatman, GE Healthcare, sup. no. WB120055) for storage and transport. FTA cards were individually bagged with desiccant and placed in a larger bag or container with additional desiccant for transportation. Researchers collected an environmental sample, dirt adjacent fecal deposits, on FTA cards as a negative control from each site to use for downstream filtering purposes.

The first set of samples came from a shelter in Hyderabad, India, in collaboration with the Blue Cross of Hyderabad. In the summer of 2019, researchers collected fecal samples non-invasively from 20 dogs at the shelter that had not been on antibiotics within the last year, also recording their sex and age when known. In addition, researchers collected fecal samples opportunistically from 14 dogs brought into the shelter from the streets as part of routine operations; samples were collected within 2–3 h of arrival. Fecal samples were collected from the centre of each deposit. The shelter dogs were fed a mixed diet of rice, lentils, yogurt and dog food. The diet of the stray dogs was unknown, but they likely scavenged from human leftovers.

The second population came from Long Nguapha village in Houaphanh Province, Laos; here, researchers collected fecal samples opportunistically from 28 dogs near the village in the winter of 2018. Fecal samples were collected from the centre of deposits to minimize contaminants. These dogs have been observed to consume local agricultural products, including soya beans, maize, cassava, bamboo, corn, sticky rice, sorghum and fish from nearby rivers [74].

Finally, pet owners in Rustenburg, Pretoria and Johannesburg in South Africa were approached to collaborate on this project in the fall of 2019. Pet owners who agreed to the study opportunistically collected fecal samples from their dogs and completed surveys that asked for information regarding pet age, breed, sex, diet, health status and exposure to other dogs or animals. Dogs were listed as living in urban, suburban or farm environments. Pet owners were also given the opportunity to opt-in for personalized reports on the microbiomes of their pets, in addition to the general findings of the study. Nineteen dogs were sampled from this population.

For comparison purposes, four beagles at the University of Illinois, Urbana–Champaign were also sampled (IACUC no. 19174). These dogs were fed a dry kibble diet (Purina Dog Chow, Nestlé Purina PetCare Company, St Louis, MO) for two weeks leading up to sample collection in cryovials and storage at −80°C. As an intra-group control comparison between diet treatments, these dogs were then transitioned to a Laotian imitation diet, reflecting the known diet of the dog population from Laos sampled in this study [74]. This diet included cassava, sorghum grains, bamboo shoots, maize, sticky rice, catfish and soya beans (electronic supplementary material, table S2). After consuming this diet for two weeks, fecal samples were collected in cryovials and stored at −80°C.

The coprolite samples used in this study come from data produced by Witt et al. [75], which in turn also includes comparative modern fecal microbiome data from Algya et al. [67] split into 'HP' and 'LP' groups (high-protein and low-protein). In short, the coprolites belong to a population of dogs from the Janey B. Goode archaeological site in southern Illinois, dating back to the Late Woodland and Terminal Late Woodland periods, approximately 1000 years ago [61]. As noted in these studies, these dogs likely assisted in carrying materials for the humans at the site, were interred within a village near houses and appear to have eaten a mix of wild and domesticated plants and fish based on macroscopic, isotopic and genetic analyses [61,75].

### (b) DNA extraction and sequencing

DNA extraction and library preparation took place in a designated pre-PCR laboratory space. For microbial DNA extraction, we used the Qiagen DNA PowerSoil kit (Qiagen, cat. no. 47014), as per the standard protocol, except the initial process as noted here. FTA cards were cut into strips, with the scissors being bleached and cleaned with ethanol to remove any bleach residue in between cuts. Strips were placed into the bead tubes for vortexing, while avoiding packing tubes to the point that prevented proper mixing of the beads and strips. To allow for proper mixing without losing samples, we conducted multiple extractions from each individual's duplicate (repeat extractions) and then pooled each individual's repeat extractions after eluting from the final filter. For contamination control, we ran each round of extractions (8–24 samples) with a negative (blank) tube; upon verifying negligible DNA concentration with Qubit, these were pooled for library preparation.

Resulting DNA extracts were quantified on the Qubit HS dsDNA platform (ThermoFisher Scientific, cat. no. Q32851). The Illumina DNA Prep library kit (Illumina, cat. no. 20018705) was used to prepare high-throughput libraries of all samples and approximately 20% of duplicates, following the standard protocol. Site-specific negatives, as well as extraction and library negatives, were also sequenced. We randomly selected duplicates for sequencing to ensure consistency, and pooled samples that did not have their duplicate sequenced separately. This resulted in 89 unique individuals, 18 duplicates and 6 negatives, which were sequenced on the NovaSeq 6000 (Illumina) on a single S4 lane, with 2 × 150 bp paired-end reads at the Core Sequencing Facility at the University of Illinois Urbana–Champaign.

The coprolite samples were extracted and sequenced as described in Witt et al. [75]. Important changes from the protocol observed for the above samples include the usage of a designated ancient DNA laboratory for sample processing, as well as an ancient DNA laboratory pipeline for DNA extraction, as opposed to the more conventional pipeline described above. The library preparation methods and sequencing platform were also different, using the KAPA Library Preparation Kits (Roche) and the Hi-Seq 4000 (Illumina) to produce 100 bp single-end reads. The modern comparative samples from Witt et al. [75] were likewise included for analysis here; their extraction and sequencing

procedures are described by Witt *et al.* [75]; other than being processed in a modern pre-PCR laboratory space and without the ancient DNA specific alterations to DNA extraction, the library preparation and sequencing methods are consistent with the coprolites [75].

## (c) Bioinformatic analyses

To assess paired-end read quality, we used FastQC [76]; only the negatives from the sites and laboratory showed high levels of dimerization. KneadData [77] was used to construct a site-specific filtering database, including the human genome, the negative from each site, and reads from the extraction and library negatives. All samples were filtered against the generated database using the default parameters. To identify taxonomic alignment, we then profiled samples using MetaPhlAn 3 [77], combining forward and reverse reads into a single file and keeping sample duplicates separate (electronic supplementary material, tables S1 and S3). Initial analyses of duplicate variability suggested that they were highly similar and occasionally had low counts (less than 300) of taxa that were only present in one duplicate. Duplicates were then combined and re-run through the above steps as a single file.

To estimate matches in the coprolites to sampled populations, we used SourceTracker [78,79] on the populations in this study and the coprolites from Witt *et al.* [75]. Populations were grouped as designated in the electronic supplementary material, table S3; the modern dogs used for comparison in Witt *et al.* were also included, as they were sequenced in a manner more similar to the coprolites [67,75]. The SourceTracker pipeline was run with default parameters, with an alpha value of 0.01 manually entered, rather than calculated. Taxonomic profiles from MetaPhlAn 3 were imported into R, using the phyloseq package [80,81] for calculating alpha diversity and beta diversity measures, with ggplot2 used for graphics (electronic supplementary material, table S4) [79]. To account for variation in read counts, we used the DESeq2 package in R [82], which provides a variance-stabilized transformation that was used for principal coordinate analysis (PCoA) and taxonomic differential abundance (electronic supplementary material, table S5) [79,81]. To measure sources of variation in the data, PERMANOVA was applied through the vegan package in R [83], using 50 000 permutations. To identify functional alignments in samples, we used HUMANN3 [77]. Profiles were normalized to counts-per-million and organized by enzyme classes. Functional novelty, defined as gene content unable to be aligned to the UniRef gene catalogue, was assessed via PPANINI [77,79].

To better represent the diversity and variation within US dogs, we included published data from Coelho *et al.* processed in a comparable manner to this study's sequence data [24]. In short, shotgun sequences were retrieved from the European nucleotide archive under Project PRJEB20308 and compiled. We trimmed and filtered the compiled sequences with the same KneadData database used on the US dog data above, concatenated the paired-end reads and then profiled the resulting samples with MetaPhlAn 3 [77]. This resulted in 31 additional individual dogs, 23 of which had duplicate data from a dietary experiment involving either a high-protein low-carbohydrate (HPLC) or low-protein high-carbohydrate (LPHC) diet compared to baseline, as described in the original study [24].

## 3. Results

### (a) Global canine microbiomes

Chao1 alpha diversity measures, as well as a Kruskal–Wallis test and subsequent paired Wilcoxon test indicate that when grouped by country, the dogs from Laos are significantly more diverse than populations from other countries, while the dogs from these other countries were relatively similar

in alpha diversity scores (figure 1; electronic supplementary material, figure S1; Kruskal–Wallis test, $p < 0.0001$, electronic supplementary material, table S4). PERMANOVA was used to identify sources of variation in microbial taxonomy across populations, taking into account each dog's country, local environment where possible, sex, diet, individuals and the sequencing depth. Of the listed factors, individuals described the most variation (40%, pseudo-$F = 4.59$), while country (27%, pseudo-$F = 32.76$), local environment (7%, pseudo-$F = 8.06$), sex (4%, pseudo-$F = 4.98$), diet (4%, pseudo-$F = 3.5$) and sequencing depth (2%, pseudo-$F = 4.13$) all described lower amounts. Many taxa not frequently observed in previously published canine microbiomes were observed across the novel populations, as listed in the electronic supplementary material, table S5.

### (b) Canine microbiomes over time

SourceTracker analysis of the coprolites suggests higher similarity to the dogs from Laos than the US dogs used in the original study (figure 2). While unknown and contaminant partitions are still observed, they are overall reduced from the original study, which only included the 'LP' and 'HP' (low-protein and high-protein) US dogs for comparison. While the dogs from India, South Africa and the US from the dietary experiment in this study were included in the analysis, no partitions were assigned to them from any coprolite. The coprolites and dogs from Laos are both marked by an increased abundance of *Enterococcus* and *Lactococcus* compared to other canine microbiomes. *Cellulosimicrobium cellulans,* a potential pathogen, was present in both the coprolites and dogs from Laos.

Chao1 alpha diversity indicates that the coprolites vary greatly in alpha diversity, though the coprolites also include the samples with the lowest measures (figure 1). Notably, the coprolites are uniformly depleted in Bacteroidetes compared to other populations and have a number of putative soil taxa, like *Saccharamonospora azurea* and *Staphylococcus albus*, as well as potential pathogens like *Rhodococcus hoagii*. *Turicibacter sanguinis* is uniquely present in JBGC16, but none of the other coprolites. This taxon also appears in samples collected from dogs from the US, Laos and India. The JBGC16 coprolite clusters independently from other coprolites in PCoA; it contains taxa that match to many of the non-US populations, whereas the remaining coprolites are largely separated and more similar to soil and one dog from Laos (figure 3; electronic supplementary material, figure S2). This trend was consistent even with the addition of a more diverse US dog population (electronic supplementary material, figure S4A,B) [24].

### (c) Diet versus environment

Taxonomically, the shift in diet altered the microbial profiles of the sampled beagles, increasing the abundance and diversity of Proteobacteria and reducing the abundance of *Bacteroides*. Despite this change, the dogs fed the imitation diet were not taxonomically similar to the dogs from Laos, maintaining similar levels of Chao1 alpha diversity (figure 1; electronic supplementary material, figure S4A). Contrary to the taxonomic results, enzymatic pathways in the microbiomes of the dogs highlighted a great deal of similarity, hinting at functional redundancy (electronic supplementary material, figure S3). Most differences that exist between these populations are due to taxa-specific

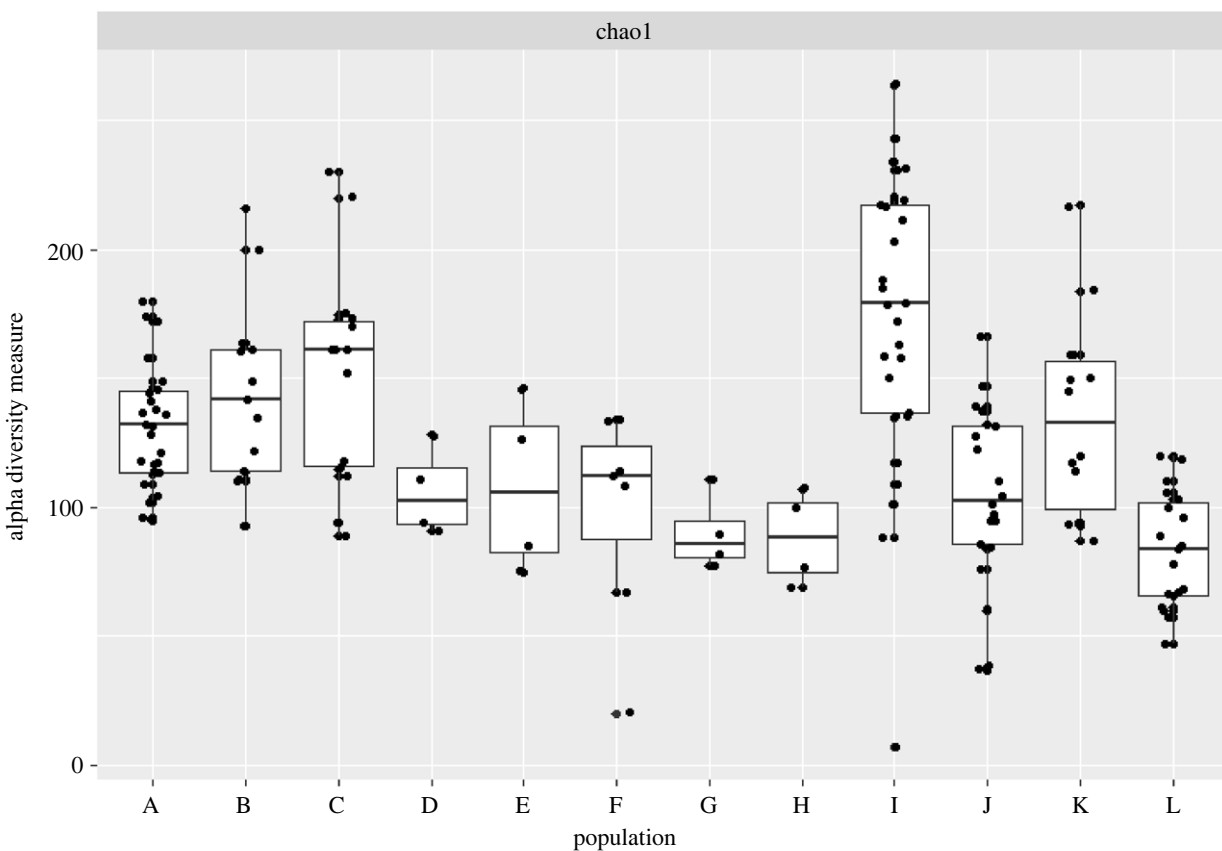

**Figure 1.** Chao1 alpha diversity of all study populations, broken down into the most discrete grouping available within countries. Populations A, B and C refer to the dogs from the US-fed baseline, HPLC or LPHC diets from the Coelho *et al.* study respectively. Populations D and E are dogs from the US-fed high- and low-protein diets, respectively, from the Witt *et al.* study [75]. Population F is the coprolites from the Witt *et al.* study [75]. Populations G and H are the dogs from the US fed the kibble and imitation diet, respectively. Population I are the dogs from Laos. Populations J and K are the dogs from India, from the shelter and strays, respectively. Population L is the dogs from South Africa. Populations are rarely significantly different from one another when separated; the dogs from South Africa are significantly less diverse than the dogs from the Coelho *et al.* study and the dogs from Laos ( $p < 0.0001$ ), and the shelter dogs from India are significantly less diverse than the dogs from Laos ( $p < 0.0001$ ).

homologues of genes, rather than entirely novel suites of functions, as is observed in the example of daidzein metabolism (electronic supplementary material, table S6). The dietary experiments, both with the imitation diet for the beagles as well as the high- and low-protein diet, are well-resolved in coordinate space and better-resolved based on functional profiles than on taxonomic data (electronic supplementary material, figure S4C). It bears notice that these results do not highlight the full extent of the functional diversity in these samples. Analysis of novel, homologous and characterized genes indicates that the coprolites include a large fraction of novel genes, meaning that there may be unknown diversity represented here (electronic supplementary material, figure S5A). Breaking down the coprolites reveals a general similarity in ratios of homologous to novel genes, except for JBGC16, which has a larger fraction of homologous and characterized genes than the other coprolites, which may be due to lower sample quality and soil contamination in other coprolites (electronic supplementary material, figure S5B).

## 4. Discussion

This study represents a novel set of shotgun microbiome data from the fecal samples of dogs in variably industrialized contexts. It includes pets from South Africa, dogs from both shelter and stray contexts in India, and dogs from a rural village in Laos. The breadth of lifestyles and diets represented in these dogs is a novel addition to the existing literature on canine fecal microbiomes and allows for better contextualization of previous and future results. We also demonstrate how, consistent with human studies, microbiomes from rural village dogs are more diverse than microbiomes of industrialized individuals.

The similarities observed between the best-preserved coprolite and non-US populations highlight the potential offered by studying populations engaged in lifestyles that mimic the exposures of ancient populations [54,84]. Much like the ancient dogs, the Laotian population consumes plants in both raw and cooked form, as well as fish [74]. They also do not live as pets, but rather around and in association with humans—potentially an even greater separation than what would have occurred at Janey B. Goode, given the labour the ancient dogs appear to have provided [61]. Within this context, it seems reasonable that a few of the coprolites might show better matches to the dogs from Laos than other populations, especially the coprolite that Witt *et al.* noted as the best preserved, JBGC16 (figure 2) [75]. We observe that JBGC16 appears to cluster with not just dogs from Laos, but non-US dogs more broadly. As a previous study on these coprolites noted, a distinguishing feature of the coprolites was their soil-associated taxa, including rhizobia [75]. The similarity between the non-US dogs and the coprolites is not a result of an overlap in these

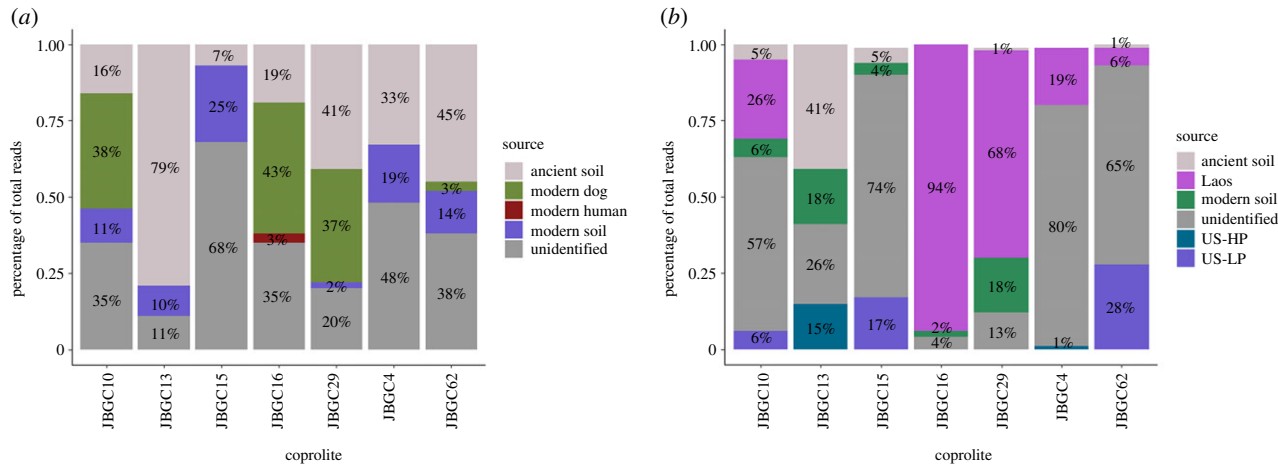

**Figure 2.** SourceTracker results when partitioning microbial results from the coprolites. (a) Original plot from Witt et al. [75] with the same coprolites re-analysed in this study, listed across the x-axis. Modern Dog reflects both the HP/LP fractions separated in (b). (b) SourceTracker results with additional populations from this study. Missing populations (dogs from India and South Africa, US dogs on the kibble and Laotian imitation diet, and US dogs fed a baseline diet from Coelho et al. [24]) were omitted due to having no assigned partitions. In comparison to (a), fractions shift noticeably to the dogs from Laos across coprolites.

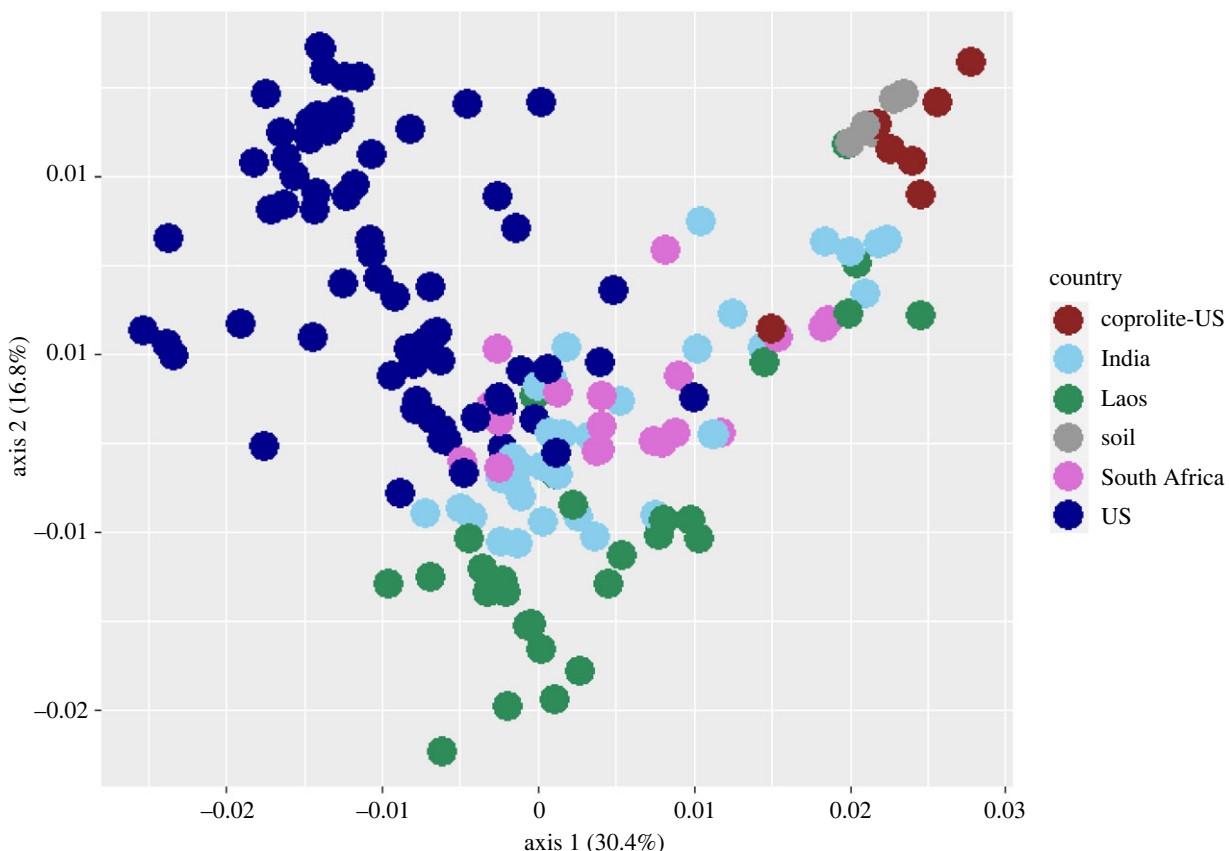

**Figure 3.** PCoA of all samples in the study, based on Bray–Curtis distances calculated from taxonomic dissimilarities. Soil refers to samples used as controls in the coprolite study. The US population is an aggregate of the US dogs from this study, the dogs from the Witt et al. paper and the dogs from the Coelho et al. study [24,67,75]. The percentage of variation explained in the dataset by each axis is provided in the brackets. Despite sequencing methodology changes between the coprolites and most of the US samples, they generally cluster by population based on taxonomic distance, though no population is specifically discrete from another.

contaminant taxa—those microbes are absent in these non-US dogs. Instead, this similarity can be attributed to the unique microbes present in these dogs and absent in other populations (electronic supplementary material, table S5). Contaminants and unknown partitions still represent large fractions of the coprolite data, but as more geographically diverse populations are sampled, these fractions will be better identified. The creation of a coprolite-specific public

database would help in this regard, by allowing for the identification of reads and taxa that may be consistent with ancient contaminants, or, more interestingly, ancient ancestors of modern-day symbionts.

The observed trend of functional redundancy is well-supported by the gross overlap in the majority of functional pathways across the populations (electronic supplementary material, figures S3 and S4C) and has been previously

identified in the literature [85]. Outside of homologues unique to various species, few enzymatic pathways were identified that were unique to one population compared to all others, like L-2-oxoglutarate carboxylase in the dogs from Laos. The relative similarity of the ratio of homologues to novel proteins further emphasizes this similarity (electronic supplementary material, figure S5). The exceptions to this trend are the coprolites, which bore a great deal of novel genes that could not be identified for this comparison, some of which are likely soil-associated and not of suitable comparison to the fecal microbiomes characterized here (electronic supplementary material, figure S5B). In keeping with previous results, among the coprolites, JBGC16 had the least number of novel genes and the greatest portion of characterized or homologous genes, though this was still greater than most other individual samples from other populations [75]. In the dogs from India, the stray population has a greater partition of novel genes compared to their shelter counterparts, perhaps reflecting their foraged diet or pathogenic influences, or both (figure 1). Both groups, stray and shelter dogs, had greater diversity in *Lactobacillus* species compared to other studied populations (electronic supplementary material, table S5). This difference may be attributed to the lactose degradation function these taxa offer, a unique metabolic function absent in most adult mammals. A single household's dogs from South Africa similarly demonstrated higher *Lactobacillus* diversity, where the dogs also consumed dairy products. These trends are further described in the supplement, but highlight another example of functional redundancy across populations.

There are limitations to the sampling and analyses presented here that should be taken into consideration when contextualizing these results. FTA cards have been used for fecal samples and produce consistent results, but they are not the gold standard of fresh-frozen samples, and will produce biases in microbiome reconstruction; however, the sample collection for this study is uniform [86]. In analysing these global populations, detailed measures were included where possible, but for many individuals, specific diets, local environments and other microbially significant exposures are unknown. This is reflected by the assessment of sources of variation in these microbiomes; individuals count for a large percentage of variation, and country overlaps with local environment and diet when these factors are unknown. In the dogs from Laos, for example, the country, local environment and diet were the same across all individuals, making it more difficult to truly assess the contributions of these factors. Seasonality, which is commonly tracked in longitudinal microbiome studies of wild populations, also overlaps with geographic populations, making it difficult to assess changes it may introduce. There are also difficulties in assigning taxonomic information to sequences from novel populations (electronic supplementary material, table S1). Despite these challenges, increased efforts to work with non-laboratory, diverse populations is essential to better understand the breadth of microbiome diversity and contextualizing our past and present results, as is shown by the novelty of the work presented here.

The concept of industrialization has been widely used in microbiome studies to encompass shifts in diet, lifestyle and exposure in ways that produce notable, consistent differences in microbial diversity across populations. As ancient microbiome studies begin to delve into this framework, it is important to recognize how these specific contexts do and do not align. While ancient and non-industrialized populations may demonstrate more similarity in microbial diversity compared to modern, industrialized populations, this result should not be conflated with the idea that modern, non-industrialized populations are ancient populations. Rather, these populations have similarities, either in diet, subsistence strategies or environmental exposure that create the conditions for the resulting observed similarity in microbial diversity—a level of nuance that should not be mistaken or ignored in contextualizing and discussing these findings.

## 5. Conclusion

This study lays the groundwork for a better understanding of human microbiome diversity in non-industrialized contexts, where it may follow the trends observed in canines in this study. Novel diversity in a Laotian dog population, for example, and the associated dietary experiment results are an indication of how the environment can alter the composition of the microbiome, similar to previously observed patterns in human microbiomes. While there is a breadth of diversity across the dog microbiomes in this study, it is worth noting the gradient of overlap that exists, suggesting some consistency in taxonomy across populations. Furthermore, the ability for multiple, varying taxa to demonstrate a functional convergence indicates that despite this global taxonomic diversity, there are multiple paths to similar outcomes in metabolism. Finally, we highlight the importance of modern analogues for ancient studies, as it only improves our ability to discern the validity of ancient microbiomes as more ancient samples are sequenced.

Ethics. This work was approved by the University of Illinois Urbana–Champaign Institutional Animal Care and Use Committee (IACUC no. 19174).

Data accessibility. The scripts and code are available at (https://github.com/kyarlagadda/shotgun_microbiome_pipeline/tree/main). Sequence data is freely available by contacting the corresponding author, and through SRA under Project ID PRJNA738608 (https://www.ncbi.nlm.nih.gov/bioproject/PRJNA738608).

The metadata are provided in the electronic supplementary material [87].

Authors' contributions. K.Y.: conceptualization, data curation, formal analysis, investigation, methodology, project administration, resources, software, supervision, validation, visualization, writing—original draft and writing—review and editing; A.J.Z.: data curation, investigation, resources and writing—review and editing; A.d.F.: data curation, investigation, methodology, project administration, resources, supervision and writing—review and editing; T.P.: data curation, investigation, methodology, project administration, resources, supervision and writing—review and editing; A.G.R.-C.: data curation, software and writing—review and editing; C.R.: formal analysis, methodology and writing—review and editing; L.S.: data curation, investigation, methodology, project administration, resources and writing—review and editing; K.S.S.: conceptualization, funding acquisition, methodology, project administration, resources, supervision, validation and writing—review and editing; R.S.M.: conceptualization, funding acquisition, methodology, project administration, resources, supervision, validation and writing—review and editing.

All authors gave final approval for publication and agreed to be held accountable for the work performed therein.

Conflict of interest declaration. We declare we have no competing interests.

Funding. This study was funded by USDA (grant no. ILLU-538-937) and University of Illinois at Urbana–Champaign.

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
