## [Peer Review File · Proceedings of the Royal Society B: Biological Sciences]

Review History

RSPB-2021-1367.R0 (Original submission)

Review form: Reviewer 1

Recommendation

Major revision is needed (please make suggestions in comments)

Scientific importance: Is the manuscript an original and important contribution to its field?

Excellent

General interest: Is the paper of sufficient general interest?

Excellent

Quality of the paper: Is the overall quality of the paper suitable?

Acceptable

Is the length of the paper justified?

Yes

Should the paper be seen by a specialist statistical reviewer?

No

Do you have any concerns about statistical analyses in this paper? If so, please specify them explicitly in your report.

No

It is a condition of publication that authors make their supporting data, code and materials available - either as supplementary material or hosted in an external repository. Please rate, if applicable, the supporting data on the following criteria.

Is it accessible?

Yes

Is it clear?

Yes

Is it adequate?

Yes

Do you have any ethical concerns with this paper?

No

Comments to the Author

The paper draws on novel and diverse dog gut microbiome samples to improve our broader understanding of spatial and temporal patterns in the dog microbiome, with particular attention to diet shifts associated with human industrialization. We think this is an interesting paper with a lot of potential to contribute to the field in meaningful ways. However, we suggest revising the structure to simplify and strengthen the message. The overall questions appear to be about diet and human industrialization and their effects on the microbiome – this theme could be brought out more centrally in the introduction. The paper includes three different ways to look at these dynamics – spatially across populations, temporally with coprolites, and temporally with controlled lab experiments. As written, these approaches are described and then their utility for testing questions of diet/industrialization is described. We suggest that diet/industrialization could be first highlighted in the introduction and then each of the three approaches, and associated gaps in the literature could be described. The methods and results could also be formatted into three sections to reflect this structure and move more systematically through the data. This format would underscore the authors' points about improving the broader context within which datasets are being analyzed and interpreted while simultaneously allowing readers to more explicitly compare the different approaches and how they complement one another. This is just one option, and there may be others, but we think that the true power and contribution of this dataset is currently not as clear as it could be. Finally, we think the wealth of functional data generated by the authors could be better leveraged (i.e. there is no main figure addressing the functional data and it is only discussed in broad terms). Detailed comments follow:

Introduction:

Line 28, Are you referring to numerous dog studies or studies in general?

Line 31, All dogs live in human contexts – are you maybe referring to dogs that are feral? It may also be prudent to briefly discuss the difference between wild dog and a feral domesticated dog (genetics vs. the environment).

It has been shown that the dog microbiome is relatively less diverse than other pets, like cats – is that true or because the data we have is skewed to Western industrial pet contexts?

Line 37: When introducing coprolites, there should be a clear connection about the hypothesis that microbiomes from coprolites may be more similar to dog microbiomes from non-Western and non-industrialized contexts. It is briefly mentioned in Lines 54-58, but it should be discussed in more detail. It will be easier for readers to understand why having better modern sampling in the context of coprolite data with this information.

Line 54, Modern industrialized human populations?

Methods:

Lines 77-78, Is the environmental sample a swab sample or on an FTA card? Please elaborate.

Since the samples for each population were collected at different times, and thus seasons, did the authors test if there was a seasonal effect on microbial composition and functional pathways?

Was there no survey data for dogs from India and Laos?

Was there a control group when beagles were transitioned to an imitation Laotian diet? Since the other US dogs did not get Laotian diet, how well can they be used as a control for the diet experiment?

To our understanding, you have samples from pets in SA, strays and shelter dogs in India, dogs from a rural village in Laos, pets/lab dogs in US + Witt and Coelho dogs in the US, and dog coprolites from the US. This means you just have one representative group from each geographic location – how will you tell the difference between the effect of geography and the local environment (differences in levels of human interaction, diet, pathogen pressure) on the dog microbiome? It is highly likely that the effect of geography will mask the effect of local environment in statistical analyses. We suggest a better explanation or discussion of the local environment per population. Additionally, can you take publicly available data (i.e., pets in India or strays in SA) and include it for more robust analysis, just like you did with the Coelho study? Breed has previously been shown to vary with microbial composition. Did the authors analyze the breed effect in the dogs from SA? Most likely there won't be information about breeds in stray and shelter dogs.

Lines 127-128, Why did the authors sequence duplicates?

Line 136, Are the modern comparative samples the Coelho or the Witt samples?

Line 141: Were the sequences all trimmed to the same length? It looks like different sequencing platforms resulted in different sequence lengths.

Line 151: Why use Source Tracker on different populations than those described earlier? Also, the goal of using SourceTracker could be described more clearly ('partitions of sources' is a little jargon).

Line 163: Authors should also map to CAZ (for diet) or other functional databases for a better functional readout.

Results:

Line 179, Figure designation is missing and the same results seem to be repeated in these first two sentences.

Line 183, Present test statistics for linear model

Line 184: Linear model analyzed effect of country, local environment (urban, suburban, rural – was not discussed/detailed beforehand), sex, diet, and sequencing depth. The authors found that diet explained much less of microbial composition compared to geography, but isn't diet and local environment confounded by geography because each population has singular diets and local environments?

There should be more functional figures than just a supplemental table about daidzein metabolism.

How did the authors tease apart which taxa were of interest/differentially abundant?

How were the survey data from the SA dogs incorporated into models?

Did the authors include individuals as a random effect in the linear models? This is important since some dogs were sampled twice.

Discussion:

The authors should provide more discussion about the diet experiment and if it follows trends from other canine diet experiments (i.e., Reese et al., 2021 – diet experiment with dogs and wolves).

The authors should also provide more discussion on why India and SA dogs were more similar to US dogs compared to Laos.

Please discuss the limitations with FTA cards as well as limitations in general.

Lines 254-255, which enzymatic pathways were unique?

Figures:

Figure 1: It seems reasonable to include Fig. S1 in the main text instead of Fig. 1 (alpha diversity

of all populations).

Figure 2: This is a little difficult to look at. Although the authors want to indicate the utility of the Laos data in particular, it might help to make the categories the same in both plots (modern dog, modern human, modern soil, etc.). At the very least, the categories that are shared between panels should use the same color.

Figure 3: The authors should include Figure S2 (PCoA of all the populations) in the main text and move this to supplemental. Also, the legend is confusing. Are the coprolites included twice? Once by themselves and then another time "The US dogs include those across all dietary treatments in this study, the coprolite study, and additional individuals across dietary treatments in the Coelho et al. study"? What beta diversity metric is the PCoA based on?

Review form: Reviewer 2

Recommendation

Major revision is needed (please make suggestions in comments)

Scientific importance: Is the manuscript an original and important contribution to its field?

Good

General interest: Is the paper of sufficient general interest?

Excellent

Quality of the paper: Is the overall quality of the paper suitable?

Acceptable

Is the length of the paper justified?

Yes

Should the paper be seen by a specialist statistical reviewer?

No

Do you have any concerns about statistical analyses in this paper? If so, please specify them explicitly in your report.

Yes

It is a condition of publication that authors make their supporting data, code and materials available - either as supplementary material or hosted in an external repository. Please rate, if applicable, the supporting data on the following criteria.

Is it accessible?

Yes

Is it clear?

No

Is it adequate?

No

Do you have any ethical concerns with this paper?

No

Comments to the Author

The authors have conducted an assessment of gut microbial taxonomy and function for several populations of dogs. They present multiple new datasets, including urban shelter dogs from India, non-pet human-associated dogs from an agricultural community in Laos, pet dogs from South Africa, and beagles that were transitioned onto a diet similar in composition to that of the agricultural dogs. Additionally, they include previously published data from dogs fed high/low protein diets, and ancient DNA from 5 coprolites, where the dogs are presumed to have consumed a mix of wild and domesticated foods. The authors used shotgun metagenomic sequencing followed by taxonomic profiling, functional profiling, and the identification of microbial “source” populations to assess the relationship modern and ancient dog gut microbiota.

The authors present an outstanding sample set, and I think there is real promise for a substantial contribution to the literature. The increased diversity in the Laotian dogs, is an exciting result, and it’s especially interesting to see that there is a potential overlap between microbial taxa in the coprolites and this non-pet population. However, I have some concerns about your analysis, particularly how you chose to present your results.

1) Generally speaking, I think you need to include more information in your methods and results sections. The text and degree of explanation about how you conducted your analysis is rather compressed. In particular, it would be good if you could provide more detail in your bioinformatics/data analysis section. For example, there are only 3 lines dedicated to the entire functional pipeline. If this is running up against the length limitation, I suggest you include further details in a supplemental methods/results section. If that is not possible, could you provide reproducible code? I checked your github link, but there is no reproducible code there, just a list of tools with descriptions of what they do in a general sense.

2) It is quite common to find different clustering patterns depending on the normalization/filtration/distance/ordination choices you make. This is of particular concern to me, because your coprolite analysis is heavily dependent on the placement of a single point, coprolite JBGC16, (figures 3, S2A). The ordination in Figure S2A shows that JBGC16 overlaps with samples from India, Laos, and South Africa, whereas the ordination in Figure 3 excludes multiple populations, giving what seems to me an unfair impression of the similarity between the Laotian samples and the coprolites to the exclusion of the non-agrarian populations. If I am misunderstanding something here, please clarify how Figure S2A is consistent with your statement on lines 204-5, “...with the coprolites appearing more similar to the dogs from Laos in coordinate space, regardless of diet.”

Furthermore, please provide more detail in your methods section about the generation of your beta diversity ordinations. For example, did you filter rare taxa prior to beta diversity analysis? Also, you mention using the DESeq variance stabilizing transformation to normalize your ASV counts. DESeq VST is great for differential abundance testing, but if you look at your transformed ASV counts, I expect that you will see a fair number of negative values. Did you do anything to adjust for this? As I understand it, Bray-Curtis distance is not robust to negative values, so if that’s the case (which it often is) you should use a different distance metric or normalization approach for your ordinations. If you prefer to stick with Bray-Curtis, you should probably use a relative abundance normalization, otherwise weighted unfrac is a better choice for the VST normalization. Take a look at figure 4 in McMurdie and Holmes (2014), which illustrates the relationship between clustering accuracy, effect size and normalization methods (written by the authors of phyloseq, who are the main proponents of using the VST). However you decide to go about this, please justify your choice and provide greater detail in the paper.

3) I think that your alpha diversity analysis needs further explanation. You describe some populations as “relatively similar.” Are they significantly different? Did you only test the Laotian samples with pairwise tests or did you test all pairs of populations? It would be helpful to include figure S1 along with figure 1 in the main paper to get a sense of the full diversity of these

populations prior to lumping them together. Also, please clearly show on the figures which box plots are significantly different in your paired Wilcoxon tests (or include this information in a supplemental table).

Secondly, your first mention of a linear model to identify sources of variation across populations is in the results section. You need to describe precisely how this model was built in the methods. Also, how were the amounts of variation calculated? Was there a model testing framework? What software package did you use? etc.

4) You mention finding different taxa in different populations, but I don't see much in the way of quantitative presentation of relative and differential abundances of microbial taxa in your samples. It's hard to tell how meaningful some of your interpretations are without including these results. For example, the caption for table S3 reads "Notable rare taxa or taxa of interest observed in novel study populations. Listed taxa were often differentially abundant in the listed population." Did you find these taxa of interest one time? Or were they common?

5) In the functional results section, you write, " ...the coprolites alone stand out as a single population cluster (Figure S2B)." I don't see how this is the case when the coprolite points in this figure overlap with points from India, Laos, South Africa, and soil.

6) Could you provide further justification/explanation for the use of source tracker? Of course, I understand that you are not making this literal connection, but none of the modern samples are actual sources of the microbes in the coprolites. Perhaps I am wrong here, but I always presumed that source tracker was designed with the expectation of potential transmission between sources and sinks. Again, I might be completely wrong here, so please correct me if am!

7) Could you include a supplemental table with details of the different diets? What proportions of these foods are consumed? Can you estimate macronutrient contents?

McMurdie PJ and Holmes S (2014) "Waste not want not: why rarefying microbiome data is inadmissible. PLOS Computational Biology. 10(4)

Decision letter (RSPB-2021-1367.R0)

02-Aug-2021

Dear Mr Yarlagadda:

I am writing to inform you that your manuscript RSPB-2021-1367 entitled "Non-Western Canid Sampling Provides Novel Insights into Pre-Industrial Microbiomes" has, in its current form, been rejected for publication in Proceedings B.

This action has been taken on the advice of referees, who have recommended that substantial revisions are necessary. With this in mind we would be happy to consider a resubmission, provided the comments of the referees are fully addressed. However please note that this is not a provisional acceptance.

Sincerely,
Dr Sasha Dall
<mailto:proceedingsb@royalsociety.org>

Associate Editor

Board Member: 1

Comments to Author:

Thank you for your submission. Your manuscript has now been assessed by two expert reviewers. I agree with their assessment that your dataset is exceptional and interesting and that you are addressing questions of broad interest. However, the reviewers also raise several important considerations that should be addressed. In particular, I would like to draw attention to the requests for more explicit hypotheses and restructuring of the introduction, and the need for more detail in the methods, including fully explained and reproducible code that others may use to assess and reconstruct your analysis pathways. I look forward to reassessing a resubmission, if you are able to address the constructive comments provided.

Reviewer(s)' Comments to Author:

Referee: 1

Comments to the Author(s)

The paper draws on novel and diverse dog gut microbiome samples to improve our broader understanding of spatial and temporal patterns in the dog microbiome, with particular attention to diet shifts associated with human industrialization. We think this is an interesting paper with a lot of potential to contribute to the field in meaningful ways. However, we suggest revising the structure to simplify and strengthen the message. The overall questions appear to be about diet and human industrialization and their effects on the microbiome – this theme could be brought out more centrally in the introduction. The paper includes three different ways to look at these dynamics – spatially across populations, temporally with coprolites, and temporally with controlled lab experiments. As written, these approaches are described and then their utility for testing questions of diet/industrialization is described. We suggest that diet/industrialization could be first highlighted in the introduction and then each of the three approaches, and associated gaps in the literature could be described. The methods and results could also be formatted into three sections to reflect this structure and move more systematically through the data. This format would underscore the authors' points about improving the broader context within which datasets are being analyzed and interpreted while simultaneously allowing readers to more explicitly compare the different approaches and how they complement one another. This is just one option, and there may be others, but we think that the true power and contribution of this dataset is currently not as clear as it could be. Finally, we think the wealth of functional data

generated by the authors could be better leveraged (i.e. there is no main figure addressing the functional data and it is only discussed in broad terms). Detailed comments follow:

Introduction:

Line 28, Are you referring to numerous dog studies or studies in general?

Line 31, All dogs live in human contexts – are you maybe referring to dogs that are feral? It may also be prudent to briefly discuss the difference between wild dog and a feral domesticated dog (genetics vs. the environment).

It has been shown that the dog microbiome is relatively less diverse than other pets, like cats – is that true or because the data we have is skewed to Western industrial pet contexts?

Line 37: When introducing coprolites, there should be a clear connection about the hypothesis that microbiomes from coprolites may be more similar to dog microbiomes from non-Western and non-industrialized contexts. It is briefly mentioned in Lines 54-58, but it should be discussed in more detail. It will be easier for readers to understand why having better modern sampling in the context of coprolite data with this information.

Line 54, Modern industrialized human populations?

Methods:

Lines 77-78, Is the environmental sample a swab sample or on an FTA card? Please elaborate.

Since the samples for each population were collected at different times, and thus seasons, did the authors test if there was a seasonal effect on microbial composition and functional pathways?

Was there no survey data for dogs from India and Laos?

Was there a control group when beagles were transitioned to an imitation Laotian diet? Since the other US dogs did not get Laotian diet, how well can they be used as a control for the diet experiment?

To our understanding, you have samples from pets in SA, strays and shelter dogs in India, dogs from a rural village in Laos, pets/lab dogs in US + Witt and Coelho dogs in the US, and dog coprolites from the US. This means you just have one representative group from each geographic location – how will you tell the difference between the effect of geography and the local environment (differences in levels of human interaction, diet, pathogen pressure) on the dog microbiome? It is highly likely that the effect of geography will mask the effect of local environment in statistical analyses. We suggest a better explanation or discussion of the local environment per population. Additionally, can you take publicly available data (i.e., pets in India or strays in SA) and include it for more robust analysis, just like you did with the Coelho study? Breed has previously been shown to vary with microbial composition. Did the authors analyze the breed effect in the dogs from SA? Most likely there won't be information about breeds in stray and shelter dogs.

Lines 127-128, Why did the authors sequence duplicates?

Line 136, Are the modern comparative samples the Coelho or the Witt samples?

Line 141: Were the sequences all trimmed to the same length? It looks like different sequencing platforms resulted in different sequence lengths.

Line 151: Why use Source Tracker on different populations than those described earlier? Also, the goal of using SourceTracker could be described more clearly ('partitions of sources' is a little jargon).

Line 163: Authors should also map to CAZ (for diet) or other functional databases for a better functional readout.

Results:

Line 179, Figure designation is missing and the same results seem to be repeated in these first two sentences.

Line 183, Present test statistics for linear model

Line 184: Linear model analyzed effect of country, local environment (urban, suburban, rural – was not discussed/detailed beforehand), sex, diet, and sequencing depth. The authors found that diet explained much less of microbial composition compared to geography, but isn't diet and local environment confounded by geography because each population has singular diets and local environments?

There should be more functional figures than just a supplemental table about daidzein metabolism.

How did the authors tease apart which taxa were of interest/differentially abundant?

How were the survey data from the SA dogs incorporated into models?

Did the authors include individuals as a random effect in the linear models? This is important since some dogs were sampled twice.

Discussion:

The authors should provide more discussion about the diet experiment and if it follows trends from other canine diet experiments (i.e., Reese et al., 2021 – diet experiment with dogs and wolves).

The authors should also provide more discussion on why India and SA dogs were more similar to US dogs compared to Laos.

Please discuss the limitations with FTA cards as well as limitations in general.

Lines 254-255, which enzymatic pathways were unique?

Figures:

Figure 1: It seems reasonable to include Fig. S1 in the main text instead of Fig. 1 (alpha diversity of all populations).

Figure 2: This is a little difficult to look at. Although the authors want to indicate the utility of the Laos data in particular, it might help to make the categories the same in both plots (modern dog, modern human, modern soil, etc.). At the very least, the categories that are shared between panels should use the same color.

Figure 3: The authors should include Figure S2 (PCoA of all the populations) in the main text and move this to supplemental. Also, the legend is confusing. Are the coprolites included twice? Once by themselves and then another time “The US dogs include those across all dietary treatments in this study, the coprolite study, and additional individuals across dietary treatments in the Coelho et al. study”? What beta diversity metric is the PCoA based on?

Referee: 2

Comments to the Author(s)

The authors have conducted an assessment of gut microbial taxonomy and function for several populations of dogs. They present multiple new datasets, including urban shelter dogs from India, non-pet human-associated dogs from an agricultural community in Laos, pet dogs from South Africa, and beagles that were transitioned onto a diet similar in composition to that of the agricultural dogs. Additionally, they include previously published data from dogs fed high/low protein diets, and ancient DNA from 5 coprolites, where the dogs are presumed to have consumed a mix of wild and domesticated foods. The authors used shotgun metagenomic sequencing followed by taxonomic profiling, functional profiling, and the identification of microbial “source” populations to assess the relationship modern and ancient dog gut microbiota.

The authors present an outstanding sample set, and I think there is real promise for a substantial contribution to the literature. The increased diversity in the Laotian dogs, is an exciting result, and it’s especially interesting to see that there is a potential overlap between microbial taxa in the coprolites and this non-pet population. However, I have some concerns about your analysis, particularly how you chose to present your results.

1) Generally speaking, I think you need to include more information in your methods and results sections. The text and degree of explanation about how you conducted your analysis is rather compressed. In particular, it would be good if you could provide more detail in your bioinformatics/data analysis section. For example, there are only 3 lines dedicated to the entire functional pipeline. If this is running up against the length limitation, I suggest you include further details in a supplemental methods/results section. If that is not possible, could you provide reproducible code? I checked your github link, but there is no reproducible code there, just a list of tools with descriptions of what they do in a general sense.

2) It is quite common to find different clustering patterns depending on the normalization/filtration/distance/ordination choices you make. This is of particular concern to

me, because your coprolite analysis is heavily dependent on the placement of a single point, coprolite JBGC16, (figures 3, S2A). The ordination in Figure S2A shows that JBGC16 overlaps with samples from India, Laos, and South Africa, whereas the ordination in Figure 3 excludes multiple populations, giving what seems to me an unfair impression of the similarity between the Laotian samples and the coprolites to the exclusion of the non-agrarian populations. If I am misunderstanding something here, please clarify how Figure S2A is consistent with your statement on lines 204-5, "...with the coprolites appearing more similar to the dogs from Laos in coordinate space, regardless of diet."

Furthermore, please provide more detail in your methods section about the generation of your beta diversity ordinations. For example, did you filter rare taxa prior to beta diversity analysis? Also, you mention using the DESeq variance stabilizing transformation to normalize your ASV counts. DESeq VST is great for differential abundance testing, but if you look at your transformed ASV counts, I expect that you will see a fair number of negative values. Did you do anything to adjust for this? As I understand it, Bray-Curtis distance is not robust to negative values, so if that's the case (which it often is) you should use a different distance metric or normalization approach for your ordinations. If you prefer to stick with Bray-Curtis, you should probably use a relative abundance normalization, otherwise weighted unifrac is a better choice for the VST normalization. Take a look at figure 4 in McMurdie and Holmes (2014), which illustrates the relationship between clustering accuracy, effect size and normalization methods (written by the authors of phyloseq, who are the main proponents of using the VST). However you decide to go about this, please justify your choice and provide greater detail in the paper.

3) I think that your alpha diversity analysis needs further explanation. You describe some populations as "relatively similar." Are they significantly different? Did you only test the Laotian samples with pairwise tests or did you test all pairs of populations? It would be helpful to include figure S1 along with figure 1 in the main paper to get a sense of the full diversity of these populations prior to lumping them together. Also, please clearly show on the figures which box plots are significantly different in your paired Wilcoxon tests (or include this information in a supplemental table).

Secondly, your first mention of a linear model to identify sources of variation across populations is in the results section. You need to describe precisely how this model was built in the methods. Also, how were the amounts of variation calculated? Was there a model testing framework? What software package did you use? etc.

4) You mention finding different taxa in different populations, but I don't see much in the way of quantitative presentation of relative and differential abundances of microbial taxa in your samples. It's hard to tell how meaningful some of your interpretations are without including these results. For example, the caption for table S3 reads "Notable rare taxa or taxa of interest observed in novel study populations. Listed taxa were often differentially abundant in the listed population." Did you find these taxa of interest one time? Or were they common?

5) In the functional results section, you write, "...the coprolites alone stand out as a single population cluster (Figure S2B)." I don't see how this is the case when the coprolite points in this figure overlap with points from India, Laos, South Africa, and soil.

6) Could you provide further justification/explanation for the use of source tracker? Of course, I understand that you are not making this literal connection, but none of the modern samples are actual sources of the microbes in the coprolites. Perhaps I am wrong here, but I always presumed that source tracker was designed with the expectation of potential transmission between sources and sinks. Again, I might be completely wrong here, so please correct me if am!

7) Could you include a supplemental table with details of the different diets? What proportions of these foods are consumed? Can you estimate macronutrient contents?

McMurdie PJ and Holmes S (2014) "Waste not want not: why rarefying microbiome data is inadmissible. PLOS Computational Biology. 10(4)

Author's Response to Decision Letter for (RSPB-2021-1367.R0)

See Appendix A.

RSPB-2022-0052.R0

Review form: Reviewer 1

Recommendation

Major revision is needed (please make suggestions in comments)

Scientific importance: Is the manuscript an original and important contribution to its field?

Excellent

General interest: Is the paper of sufficient general interest?

Excellent

Quality of the paper: Is the overall quality of the paper suitable?

Good

Is the length of the paper justified?

Yes

Should the paper be seen by a specialist statistical reviewer?

No

Do you have any concerns about statistical analyses in this paper? If so, please specify them explicitly in your report.

No

It is a condition of publication that authors make their supporting data, code and materials available - either as supplementary material or hosted in an external repository. Please rate, if applicable, the supporting data on the following criteria.

Is it accessible?

Yes

Is it clear?

Yes

Is it adequate?

Yes

Do you have any ethical concerns with this paper?

No

Comments to the Author

The authors did a commendable job incorporating previous feedback and/or defending their decisions. There are few revisions we still recommend – the line numbers refer to the track-changes version.

Introduction

- Populations can be identified more than just industrialized vs non industrialized. There are populations that live “in between” (i.e., rural populations near industrialized cities or rapidly developing populations). Indeed, the fact that the authors sample dogs that live in in-between environments (i.e., shelters) highlights this idea and should be another reason to justify using canine microbiomes as a good model for industrialization.
- In this vein, there should be a more explicit connection between the ideas of industrialization as a complex system to disentangle microbial contributions and presenting the novel canine microbiome dataset (Lines 22-30). Specifically, the authors may consider justifying why dogs are a good model to disentangle these connections.
- Missing “C” in “Canine” in Line 43
- There should be more discussion on what makes feral and strays different to pets with respect to diet, environment, and/or interactions with humans.
- Line 78, “ther”?

Methods

- Lines 163-165, how long did the researchers wait to collect the fecal sample from the 14 dogs brought into the shelter?

Results

- Line 288, what is “they” referring to – the coprolites or diversity analyses?
- While the authors have presented more detail in their metagenomics methodology, there should be more space dedicated to metagenomics results – for example, did the PERMANOVA also explain variation in the functional potential of the canine microbiomes?
- What about diet in other populations apart from beagles?

Discussion

- Much of the discussion is on the data from the Laos dogs and the coprolites – there should be more discussion on the microbiomes from other dog populations
- FTA cards preserve samples effectively, but the resulting communities differ from ethanol-preserved samples (line 299). This sentence could just use some clarification on this point (and a reference).

Figures

- Figure 3, where are the samples from SA and India in the PCoA plot?

Review form: Reviewer 2

Recommendation

Major revision is needed (please make suggestions in comments)

Scientific importance: Is the manuscript an original and important contribution to its field?

Excellent

General interest: Is the paper of sufficient general interest?

Excellent

Quality of the paper: Is the overall quality of the paper suitable?

Acceptable

Is the length of the paper justified?

Yes

Should the paper be seen by a specialist statistical reviewer?

No

Do you have any concerns about statistical analyses in this paper? If so, please specify them explicitly in your report.

Yes

It is a condition of publication that authors make their supporting data, code and materials available - either as supplementary material or hosted in an external repository. Please rate, if applicable, the supporting data on the following criteria.

Is it accessible?

Yes

Is it clear?

Yes

Is it adequate?

Yes

Do you have any ethical concerns with this paper?

No

Comments to the Author

The authors have substantially improved the paper from the first version that I read. In particular, I appreciate that they have included the differential abundance table and the expanded methods in the supplement. However, I still have concerns regarding their over-interpretation of the ordinations and justification for similarity between the coprolites and samples from Laos. I have responded to your responses to my queries in the corresponding numerical order.

1) The additional methodological details that you have provided in the expanded supplement is much improved and mostly satisfies my request.

2A) I am still having trouble with how you justify your interpretation that. “[t]he coprolites overall are more similar to the dogs from Laos.” Yes, the point you make here in your response that coprolite JBGC16 shares some taxa with samples from Laos is reasonable. However, I do not see any statistical evidence showing that the abundances of these taxa are similar in the two groups. Do you mean that they are simply present at least once in JBGC6 and at least one Laotian sample? Second, I do not agree with your use of the PCoAs (any of them) to demonstrate similarity between the ancient dogs and the dogs from Laos. Ultimately, you are relying on one sample to argue this point. I appreciate that you have softened the language in the text, but in the caption of figure S3A, you still write that, “dogs from Laos show more similarity to coprolites, with JBGC16 indicated by the red arrow.” If your argument is that the position of JCBC16 in the PCoA demonstrates that the coprolites are more similar to the samples from Laos, one could also use the single sample from Laos that clusters with the US samples to declare that all the dogs from Laos are similar to the US dogs (which is of course, false). The coprolites might well be more similar to the samples from Laos – your source tracker results show some evidence this – but your PCoAs do not. What I am saying here is that you cannot use the placement of a single coprolite in the PCoA while ignoring the placement of the rest of the coprolites to justify the similarity of the coprolites as a group to the samples from Laos as a group (especially given their positioning in Figure S2A on Axis 1, which explains over twice the variance of axis 2). Also, Figure 3 appears identical to figure S2B minus the arrow. You should replace figure 3 with Figure S2A, which shows the more fine scale differences, or include both in the manuscript.

2B) It is encouraging that the the VST transformed Bary-Curtis values are positive. However, my criticism of the use of DEseq VST with Bray-Curtis distances was two pronged, and the authors have not addressed my second point, illustrated in Figure 4 of McMurdie and Holmes (2014): in absence of a large effect size, Bray-Curtis distance is not accurate when combined with DEseq variance stabilization. Again, if you want to use the VST, you should be generating weighted unifrac distances, not Bray-Curtis distances, unless you have demonstrated a large effect size. Normally, I would not be insistent about this issue – by and large ordinations look the quite similar with different distances/transformations – but your interpretation of the similarity between the coprolites and the samples from Laos is dependent on the position of the single point, JBGC16, and as such, I don't think it is appropriate to rely on the accuracy of its positioning in the ordination when the combination of data transformation and distance metric used to generate that point has been demonstrated not to be very accurate. At the very least, I would like to see a version of figure 3 that uses weighted unifrac distance and VST or Bray-Curtis distance with relative abundance transformation.

3) I appreciate the inclusion of the additional tests, clarification of the alpha diversity analysis, and PERMANOVA. Much improved.

4) Regarding the quantitative analysis of differential abundance, I presume you mean table S5, and not S3 (or did I mix those up in my initial response? Regardless, much better, thanks. Did you also conduct this analysis on the coprolites? I don't see any reference to them. If not, how are you determining the similarity between the coprolite taxa and other populations addressed in point #2A? For example, in the manuscript, you write, "[t]he coprolites and dogs from Laos are both marked by an increased abundance of *Enterococcus* and *Lactococcus* compared to other canine microbiomes." I don't see how is this demonstrated statistically.

5) The revised text is better, but the PCoA issues discussed elsewhere need to be addressed.

6) Thank you for this explanation and the list of papers. It is very helpful.

7) Thank you for the additional diet table. Regarding the comment that you have applied the principles of McMurdie and Holmes (2014) to this paper, I don't see how that is the case, based on my comment in point #2B.

Decision letter (RSPB-2022-0052.R0)

14-Feb-2022

Dear Mr Yarlagadda:

Your manuscript has now been peer reviewed and the reviews have been assessed by an Associate Editor. The reviewers' comments (not including confidential comments to the Editor) and the comments from the Associate Editor are included at the end of this email for your reference. As you will see, the reviewers and the Editors have raised some concerns with your manuscript and we would like to invite you to revise your manuscript to address them.

Research ethics:

Use of animals and field studies:

It is a condition of publication that you make available the data and research materials supporting the results in the article (<https://royalsociety.org/journals/authors/author-guidelines/#data>). Datasets should be deposited in an appropriate publicly available repository and details of the associated accession number, link or DOI to the datasets must be included in the Data Accessibility section of the article (<https://royalsociety.org/journals/ethics-policies/data-sharing-mining/>). Reference(s) to datasets should also be included in the reference list of the article with DOIs (where available).

All supplementary materials accompanying an accepted article will be treated as in their final form. They will be published alongside the paper on the journal website and posted on the online

figshare repository. Files on figshare will be made available approximately one week before the accompanying article so that the supplementary material can be attributed a unique DOI. Please try to submit all supplementary material as a single file.

Please submit a copy of your revised paper within three weeks. If we do not hear from you within this time your manuscript will be rejected. If you are unable to meet this deadline please let us know as soon as possible, as we may be able to grant a short extension.

Best wishes,
Dr Sasha Dall
mailto:proceedingsb@royalsociety.org

Associate Editor Board Member

Comments to Author:

Both of the reviewers and I find the revised version of your manuscript much improved. We remain excited about the potential impact of this research and are highly supportive. At the same time, both of the reviewers highlight a few outstanding concerns that were not sufficiently addressed in the first revision. I agree with their assessment and draw attention in particular to the requests for revisions that would better clarify and support the conclusions drawn from the data.

Reviewer(s)' Comments to Author:

Referee: 1

Comments to the Author(s).

The authors did a commendable job incorporating previous feedback and/or defending their decisions. There are few revisions we still recommend – the line numbers refer to the track-changes version.

Introduction

- Populations can be identified more than just industrialized vs non industrialized. There are populations that live “in between” (i.e., rural populations near industrialized cities or rapidly developing populations). Indeed, the fact that the authors sample dogs that live in in-between environments (i.e., shelters) highlights this idea and should be another reason to justify using canine microbiomes as a good model for industrialization.
- In this vein, there should be a more explicit connection between the ideas of industrialization as a complex system to disentangle microbial contributions and presenting the novel canine microbiome dataset (Lines 22-30). Specifically, the authors may consider justifying why dogs are a good model to disentangle these connections.
- Missing “C” in “Canine” in Line 43
- There should be more discussion on what makes feral and strays different to pets with respect to diet, environment, and/or interactions with humans.
- Line 78, “ther”?

Methods

- Lines 163-165, how long did the researchers wait to collect the fecal sample from the 14 dogs brought into the shelter?

Results

- Line 288, what is “they” referring to – the coprolites or diversity analyses?

- While the authors have presented more detail in their metagenomics methodology, there should be more space dedicated to metagenomics results – for example, did the PERMANOVA also explain variation in the functional potential of the canine microbiomes?

- What about diet in other populations apart from beagles?

Discussion

- Much of the discussion is on the data from the Laos dogs and the coprolites – there should be more discussion on the microbiomes from other dog populations

- FTA cards preserve samples effectively, but the resulting communities differ from ethanol-preserved samples (line 299). This sentence could just use some clarification on this point (and a reference).

Figures

- Figure 3, where are the samples from SA and India in the PCoA plot?

Referee: 2

Comments to the Author(s).

The authors have substantially improved the paper from the first version that I read. In particular, I appreciate that they have included the differential abundance table and the expanded methods in the supplement. However, I still have concerns regarding their over-interpretation of the ordinations and justification for similarity between the coprolites and samples from Laos. I have responded to your responses to my queries in the corresponding numerical order.

1) The additional methodological details that you have provided in the expanded supplement is much improved and mostly satisfies my request.

2A) I am still having trouble with how you justify your interpretation that. “[t]he coprolites overall are more similar to the dogs from Laos.” Yes, the point you make here in your response that coprolite JBGC16 shares some taxa with samples from Laos is reasonable. However, I do not see any statistical evidence showing that the abundances of these taxa are similar in the two groups. Do you mean that they are simply present at least once in JBGC6 and at least one Laotian sample? Second, I do not agree with your use of the PCoAs (any of them) to demonstrate similarity between the ancient dogs and the dogs from Laos. Ultimately, you are relying on one sample to argue this point. I appreciate that you have softened the language in the text, but in the caption of figure S3A, you still write that, “dogs from Laos show more similarity to coprolites, with JBGC16 indicated by the red arrow.” If your argument is that the position of JBGC16 in the PCoA demonstrates that the coprolites are more similar to the samples from Laos, one could also use the single sample from Laos that clusters with the US samples to declare that all the dogs from Laos are similar to the US dogs (which is of course, false). The coprolites might well be more similar to the samples from Laos – your source tracker results show some evidence this – but your PCoAs do not. What I am saying here is that you cannot use the placement of a single coprolite in the PCoA while ignoring the placement of the rest of the coprolites to justify the similarity of the coprolites as a group to the samples from Laos as a group (especially given their positioning in Figure S2A on Axis 1, which explains over twice the variance of axis 2). Also, Figure 3 appears identical to figure S2B minus the arrow. You should replace figure 3 with Figure S2A, which shows the more fine scale differences, or include both in the manuscript.

2B) It is encouraging that the the VST transformed Bary-Curtis values are positive. However, my criticism of the use of DEseq VST with Bray-Curtis distances was two pronged, and the authors have not addressed my second point, illustrated in Figure 4 of McMurdie and Holmes (2014): in absence of a large effect size, Bray-Curtis distance is not accurate when combined with DEseq variance stabilization. Again, if you want to use the VST, you should be generating weighted unifrac distances, not Bray-Curtis distances, unless you have demonstrated a large effect size. Normally, I would not be insistent about this issue – by and large ordinations look the quite similar with different distances/transformations – but your interpretation of the similarity between the coprolites and the samples from Laos is dependent on the position of the single point, JBGC16, and as such, I don’t think it is appropriate to rely on the accuracy of its positioning

in the ordination when the combination of data transformation and distance metric used to generate that point has been demonstrated not to be very accurate. At the very least, I would like to see a version of figure 3 that uses weighted unifrac distance and VST or Bray-Curtis distance with relative abundance transformation.

3) I appreciate the inclusion of the additional tests, clarification of the alpha diversity analysis, and PERMANOVA. Much improved.

4) Regarding the quantitative analysis of differential abundance, I presume you mean table S5, and not S3 (or did I mix those up in my initial response? Regardless, much better, thanks. Did you also conduct this analysis on the coprolites? I don't see any reference to them. If not, how are you determining the similarity between the coprolite taxa and other populations addressed in point #2A? For example, in the manuscript, you write, "[t]he coprolites and dogs from Laos are both marked by an increased abundance of Enterococcus and Lactococcus compared to other canine microbiomes." I don't see how is this demonstrated statistically.

5) The revised text is better, but the PCoA issues discussed elsewhere need to be addressed.

6) Thank you for this explanation and the list of papers. It is very helpful.

7) Thank you for the additional diet table. Regarding the comment that you have applied the principles of McMurdie and Holmes (2014) to this paper, I don't see how that is the case, based on my comment in point #2B.

Author's Response to Decision Letter for (RSPB-2022-0052.R0)

See Appendix B.

RSPB-2022-0052.R1

Review form: Reviewer 2

Recommendation

Accept as is

Scientific importance: Is the manuscript an original and important contribution to its field?

Excellent

General interest: Is the paper of sufficient general interest?

Good

Quality of the paper: Is the overall quality of the paper suitable?

Good

Is the length of the paper justified?

Yes

Should the paper be seen by a specialist statistical reviewer?

Yes

Do you have any concerns about statistical analyses in this paper? If so, please specify them explicitly in your report.

Yes

It is a condition of publication that authors make their supporting data, code and materials available - either as supplementary material or hosted in an external repository. Please rate, if applicable, the supporting data on the following criteria.

Is it accessible?

Yes

Is it clear?

Yes

Is it adequate?

Yes

Do you have any ethical concerns with this paper?

No

Comments to the Author

Thank you for the adjustments that you have made to the manuscript. I think it has been improved. Regarding point 2B, I wrote, that "At the very least, I would like to see a version of figure 3 that uses weighted unifrac distance and VST or Bray-Curtis distance with relative abundance transformation." Thank you for explaining your case-specific issue with unifrac distances. I see your point. However, I fail to see why you disregarded my alternative recommendation that you show me a PCoA based on Bray-Curtis and relative abundances, which is certainly possible to run. Nonetheless, I will not press the point further, since you have toned down the interpretation of the PCoA, and overall it's a great paper.

Decision letter (RSPB-2022-0052.R1)

25-Mar-2022

Dear Mr Yarlagadda

I am pleased to inform you that your manuscript RSPB-2022-0052.R1 entitled "Geographically Diverse Canid Sampling Provides Novel Insights into Pre-Industrial Microbiomes" has been accepted for publication in Proceedings B.

The referee(s) have recommended publication, but also suggest some minor revisions to your manuscript. Therefore, I invite you to respond to the referee(s)' comments and revise your manuscript. Because the schedule for publication is very tight, it is a condition of publication that you submit the revised version of your manuscript within 7 days. If you do not think you will be able to meet this date please let us know.

[http://datadryad.org/submit?journalID=RSPB&manu=\(Document not available\)](http://datadryad.org/submit?journalID=RSPB&manu=(Document not available)) which will take you to your unique entry in the Dryad repository. If you have already submitted your data to dryad you can make any necessary revisions to your dataset by following the above link.

Please see <https://royalsociety.org/journals/ethics-policies/data-sharing-mining/> for more details.

Sincerely,
Dr Sasha Dall
Editor, Proceedings B
<mailto:proceedingsb@royalsociety.org>

Associate Editor:

Board Member: 1

Comments to Author:

Many thanks to the authors for submitted a revised version of their manuscript. Each time I read over this study, my enthusiasm for this interesting and timely research resurfaces. I anticipate this study will make a valuable contribution to the field.

The reviewers and myself find the revised version of the manuscript to be much improved. With one minor exception, the revisions carried out were satisfactory. However, I draw attention to a point raised by a reviewer regarding a PCoA based on Bray-Curtis and relative abundances. Please provide this in the final version ahead of publication, or provide a detailed explanation as to why this isn't possible. And as a very minor point, I also suggest the authors use a lower case "v" in the heading: Diet versus Environment.

Reviewer(s)' Comments to Author:

Referee: 2

Comments to the Author(s)

Thank you for the adjustments that you have made to the manuscript. I think it has been improved. Regarding point 2B, I wrote, that "At the very least, I would like to see a version of figure 3 that uses weighted unifrac distance and VST or Bray-Curtis distance with relative abundance transformation." Thank you for explaining your case-specific issue with unifrac distances. I see your point. However, I fail to see why you disregarded my alternative recommendation that you show me a PCoA based on Bray-Curtis and relative abundances, which is certainly possible to run. Nonetheless, I will not press the point further, since you have toned down the interpretation of the PCoA, and overall it's a great paper.

Author's Response to Decision Letter for (RSPB-2022-0052.R1)

See Appendix C.

Decision letter (RSPB-2022-0052.R2)

01-Apr-2022

Dear Mr Yarlagadda

I am pleased to inform you that your manuscript RSPB-2022-0052.R2 entitled "Geographically Diverse Canid Sampling Provides Novel Insights into Pre-Industrial Microbiomes" has been accepted for publication in Proceedings B.

The referee(s) have recommended publication, but also suggest some minor revisions to your manuscript. Therefore, I invite you to respond to the referee(s)' comments and revise your manuscript. Because the schedule for publication is very tight, it is a condition of publication that you submit the revised version of your manuscript within 7 days. If you do not think you will be able to meet this date please let us know.

In order to ensure effective and robust dissemination and appropriate credit to authors the dataset(s) used should be fully cited. To ensure archived data are available to readers, authors should include a 'data accessibility' section immediately after the acknowledgements section.

This should list the database and accession number for all data from the article that has been made publicly available, for instance:

Sincerely,
Dr Sasha Dall
Editor, Proceedings B
<mailto:proceedingsb@royalsociety.org>

Associate Editor:
Board Member
Comments to Author:

Many thanks to the authors for adding this additional supple. figure. Please note in the latest submission, the first figure caption does not say Figure 1. (There is no title). Then there are two captions called Figure 2. Please carefully review and revise the figure captions, in text citations and supplementary materials etc as needed. In the caption for Figure S4, consider "best-preserved" (rather than best preserved).

Decision letter (RSPB-2022-0052.R3)

04-Apr-2022

Dear Mr Yarlagadda

I am pleased to inform you that your manuscript entitled "Geographically Diverse Canid Sampling Provides Novel Insights into Pre-Industrial Microbiomes" has been accepted for publication in Proceedings B.

Your article has been estimated as being 9 pages long. Our Production Office will be able to confirm the exact length at proof stage.

Data Accessibility section

Open Access

Paper charges

Sincerely,

Appendix A

Associate Editor

Board Member: 1

Comments to Author:

Thank you for your submission. Your manuscript has now been assessed by two expert reviewers. I agree with their assessment that your dataset is exceptional and interesting and that you are addressing questions of broad interest. However, the reviewers also raise several important considerations that should be addressed. In particular, I would like to draw attention to the requests for more explicit hypotheses and restructuring of the introduction, and the need for more detail in the methods, including fully explained and reproducible code that others may use to assess and reconstruct your analysis pathways. I look forward to reassessing a resubmission, if you are able to address the constructive comments provided.

Reviewer(s)' Comments to Author:

Referee: 1

Comments to the Author(s)

The paper draws on novel and diverse dog gut microbiome samples to improve our broader understanding of spatial and temporal patterns in the dog microbiome, with particular attention to diet shifts associated with human industrialization. We think this is an interesting paper with a lot of potential to contribute to the field in meaningful ways. However, we suggest revising the structure to simplify and strengthen the message. The overall questions appear to be about diet and human industrialization and their effects on the microbiome – this theme could be brought out more centrally in the introduction. The paper includes three different ways to look at these dynamics – spatially across populations, temporally with coprolites, and temporally with controlled lab experiments. As written, these approaches are described and then their utility for testing questions of diet/industrialization is described. We suggest that diet/industrialization could be first highlighted in the introduction and then each of the three approaches, and associated gaps in the literature could be described. The methods and results could also be formatted into three sections to reflect this structure and move more systematically through the data. This format would underscore the authors' points about improving the broader context within which datasets are being analyzed and interpreted while simultaneously allowing readers to more explicitly compare the different approaches and how they complement one another. This is just one option, and there may be others, but we think that the true power and contribution of this dataset is currently not as clear as it could be. Finally, we think the wealth of functional data generated by the

authors could be better leveraged (i.e. there is no main figure addressing the functional data and it is only discussed in broad terms). Detailed comments follow:

We thank the reviewer for their guidance on re-organizing the introduction, and paper at large, to provide a more logical flow. We have revised the paper accordingly, creating the three groupings for the introduction and results.

Introduction:

Line 28, Are you referring to numerous dog studies or studies in general?

This line should refer to studies in general; it has been rewritten to read “For example, numerous general microbiome studies support the positive relationship between the genus *Prevotella* and dietary fiber and the genus *Bacteroides* with dietary protein – but it is unknown if these trends hold in pet populations in non-industrialized contexts.” Lines 46-49.

Line 31, All dogs live in human contexts – are you maybe referring to dogs that are feral? It may also be prudent to briefly discuss the difference between wild dog and a feral domesticated dog (genetics vs. the environment).

This is correct; we’ve expanded this sentence to include some previous studies that approach the topic, while indicating where there are still gaps: “Similarly, the lack of data on strays or feral dogs hinders our understanding of how social interaction with humans and human-associated diets influence canine microbiomes. At best, we have a few studies that compare the microbiomes of dogs and wolves, but there are known genetic changes between wolves and dogs that would affect their diet, and thus microbiome.” Lines 49-52.

It has been shown that the dog microbiome is relatively less diverse than other pets, like cats – is that true or because the data we have is skewed to Western industrial pet contexts?

We are unfamiliar with studies that might demonstrate this through a comparison of dog and cat microbiomes. Both animals have largely been studied in Western, industrial contexts as the reviewer notes, so such studies would likely have this bias. While an interesting point, the discussion of felid microbiomes is outside the scope of our study.

Line 37: When introducing coprolites, there should be a clear connection about the hypothesis that microbiomes from coprolites may be more similar to dog microbiomes from non-Western and non-industrialized contexts. It is briefly mentioned in Lines 54-58, but it should be discussed in more detail. It will be easier for readers to understand why having better modern sampling in the context of coprolite data with this information.

In re-organizing the introduction, this topic has been given its own section, and is now expanded upon as follows: “We address this problem with our first step; by sampling globally diverse populations of dogs, some pets, some strays, we have a better representation of the canine fecal microbiome. This allows us to better analyze ancient canine fecal microbiomes, with the hypothesis that these ancient microbiomes will be more similar to modern canine microbiomes from non-industrialized contexts. In fact, the dogs from Laos, who are human-associated but consume a mix of agricultural products and foraged foods, are likely to have the most similar fecal microbiomes due to their outdoor environment and mixed diet. Because industrialization would not have affected the ancient dogs, similarities between ancient and modern dog fecal microbiomes also provide insight into the question of industrialization’s impact on the microbiome. Modern dogs in industrialized contexts are expected to have less diverse and less similar microbiomes to both non-industrialized modern populations’ microbiomes and the non-industrialized ancient population’s microbiome.” Lines 79-89.

Line 54, Modern industrialized human populations?

This line originally referred to canine populations, but it has been omitted in the rewrite of the introduction.

Methods:

Lines 77-78, Is the environmental sample a swab sample or on an FTA card? Please elaborate. Since the samples for each population were collected at different times, and thus seasons, did the authors test if there was a seasonal effect on microbial composition and functional pathways?

This is an FTA card; the line has been updated to read “Researchers collected an environmental sample, dirt adjacent fecal deposits, on FTA cards as a negative control from each site to use for downstream filtering purposes.” Lines 123-125.

The samples were collected within short timeframes (<1 month) in each population. While they were different times for sampling across population, a test for a difference due to seasonality would require

more longitudinal data (to test within population changes) than we have in our dataset, as any “seasonal” differences would overlap with population differences. As requested by another comment, a section on limitations now includes an acknowledgment of this.

Was there no survey data for dogs from India and Laos?

Similar surveys to the one employed in South Africa were not employed in India and Laos due to the different contexts of these populations. In India, shelter dogs had mixed history data available, and strays had no such data. In Laos, the dogs were more akin to strays, and thus likewise lacked the regular data that the survey captured. In India, sex, diet, and health status were recorded to the best knowledge available. In Laos, individual sex was recorded when observed, but most other data was unavailable. As noted in the methods, diet was described for the whole population, rather than on an individual basis. The methods have been updated to include the partial information available in these populations.

Was there a control group when beagles were transitioned to an imitation Laotian diet? Since the other US dogs did not get Laotian diet, how well can they be used as a control for the diet experiment?

The beagles are their own controls - in this experimental design, the same four individuals are fed a kibble diet and then the Laotian diet, allowing us to control for individual variation. This is a relatively robust methodology used in many canine microbiome diet studies (see citations). If by “other US dogs” the reviewer is referring to the dogs from the Coelho et al. study, these are dogs fed a couple varieties of extruded diets that are different to the Laotian diet, and better capture the diversity of the US pet dog microbiome than our four individuals alone. The addition of these samples was intended to identify if the novel diversity observed in the microbiomes from the dogs from Laos was a result of our low sample size in our experimental population, which it did not appear to be. These additional dogs are never referred to nor used as controls; rather, they provide additional context to help support the observed differences between the dietary experimental control and treatment versus the dogs from Laos.

Algya KM, Cross TL, Leuck KN, Kastner ME, Baba T, Lye L, et al. Apparent total-tract macronutrient digestibility, serum chemistry, urinalysis, and fecal characteristics, metabolites and microbiota of adult dogs fed extruded, mildly cooked, and raw diets. *J Anim Sci.* 2018;96(9):3670-83.

Martínez-López LM, Pepper A, Pilla R, Woodward AP, Suchodolski JS, Mansfield C. Effect of sequentially fed high protein, hydrolysed protein, and high fibre diets on the faecal microbiota of healthy dogs: a cross-over study. 2020.

Alexander C, Guard, BC, Suchodolski JS, Swanson, KS. Cholestyramine decreases apparent total tract macronutrient digestibility and alters fecal characteristics and metabolites of healthy adult dogs. *J. Anim Sci.* 2019;97(3):1020-1026.

To our understanding, you have samples from pets in SA, strays and shelter dogs in India, dogs from a rural village in Laos, pets/lab dogs in US + Witt and Coelho dogs in the US, and dog coprolites from the US. This means you just have one representative group from each geographic location – how will you tell the difference between the effect of geography and the local environment (differences in levels of human interaction, diet, pathogen pressure) on the dog microbiome? It is highly likely that the effect of geography will mask the effect of local environment in statistical analyses. We suggest a better explanation or discussion of the local environment per population. Additionally, can you take publicly available data (i.e., pets in India or strays in SA) and include it for more robust analysis, just like you did with the Coelho study?

This study does not have the depth of data to discern between local environment and broad-scale geographic populations. The populations are referred to throughout as “dogs from country” to avoid the implication that these populations might be representative of whole geographic regions or countries. As noted, local environments are interesting to the questions posed here, but indistinguishable in all but two cases with the data available. For the first of these, we now reference the following in the methods for the dogs from South Africa as per the reviewer’s suggestion: “Dogs were listed as living in urban, suburban, or farm environments.” Lines 141-142.

We have added a section in the supplement further exploring the dynamics between the Indian stray and shelter dogs, the other population where we can discuss the effect of local environment. While the reviewer’s suggestion to use publicly available data is a good one, as described in our paper, there is no such comparable data to our knowledge. Canine microbiome studies are limited in geographic scope, and as the reviewer notes, this data is novel, making larger comparisons difficult.

Breed has previously been shown to vary with microbial composition. Did the authors analyze the breed effect in the dogs from SA? Most likely there won’t be information about breeds in stray and shelter dogs.

Breeds were analyzed, but there was such minimal overlap in breeds across the population that the effect of breed was indistinguishable from the individual dogs themselves. We have noted this analysis in the supplement. As the reviewer notes, this data was lacking in our other populations.

Lines 127-128, Why did the authors sequence duplicates?

Duplicate sequencing provides a useful test to ensure minimal errors in laboratory procedures that might produce biases in estimated microbiome composition in the analysis. There are a number of complications with microbiome data; it is a snapshot of an individual's microbiome, it can be difficult to capture low abundance or rare organisms, sequences not represented in the database are often ignored, and so on. Across billions of sequence reads, it is helpful to ensure that laboratory methods are consistent by sequencing duplicates and comparing them to ensure there aren't rare sequences or an abundance of artifacts in one duplicate compared to another. The referenced line now reads "We randomly selected duplicates for sequencing to ensure consistency, and pooled samples that did not have their duplicate sequenced separately." Lines 176-178.

Line 136, Are the modern comparative samples the Coelho or the Witt samples?

This refers to the Witt et al. samples. The line has been edited to reflect this more clearly: "The modern comparative samples from Witt et al. were likewise included for analysis here; their extraction and sequencing procedures are described by Witt et al." Lines 186-188.

Line 141: Were the sequences all trimmed to the same length? It looks like different sequencing platforms resulted in different sequence lengths.

Sequences were trimmed by trimmomatic based on quality scores, resulting in variable sequence length. The outputs from the two different sequencing platforms were originally different lengths (1x100bp from the Witt et al. study, 2x150bp from the newly sampled sequences in this study), but were all aligned and processed similarly after trimming. These details are now more clearly laid out in the expanded methods in the supplement.

Line 151: Why use Source Tracker on different populations than those described earlier? Also, the goal of using SourceTracker could be described more clearly ('partitions of sources' is a little jargony).

SourceTracker was used on the same populations as described earlier, simply broken into more groups where possible to provide greater clarity in the SourceTracker results (i.e., in a previous study, we had identified that grouping modern dogs vs. splitting them based on their high- and low-protein diet made a difference in SourceTracker results). The populations described earlier are still the overarching populations in the analysis. The goal has been restated as follows: “To estimate matches in the coprolites to sampled populations, we used SourceTracker on the populations in this study and the coprolites from Witt et al.” Lines 202-203

Line 163: Authors should also map to CAZ (for diet) or other functional databases for a better functional readout.

Our chosen method for alignment and protein classification utilizes the UniProt database, which maintains a broad set of annotations and was selected to account for the anticipated novelty of this dataset. Even with this, there are overlaps with the classes of enzymes CAZy aims to characterize. At a glance, the following enzymes (right) were readily found across several taxa within CAZy classes (left):

Glycoside Hydrolases - beta-galactosidases

Glycosyltransferases - fructotransferase, alpha-D-glucosyltransferase

Polysaccharide Lyases - pectin lyase, pectate lyase, hyaluronate lyase

Carbohydrate Esterases - pectinesterase, acetylxylan esterase

Auxiliary Activities - glucose-fructose oxidoreductase

Results:

Line 179, Figure designation is missing and the same results seem to be repeated in these first two sentences.

We agree with the reviewer and have condensed these lines to reflect this. The opening section now reads: “Chao1 alpha diversity measures, as well as a Kruskal-Wallis test and subsequent paired Wilcoxon test indicate that the dogs from Laos are significantly more diverse than populations from other countries, while the dogs from these other countries were relatively similar in alpha diversity scores (Figure 1; Kruskal-Wallis test, $p < 0.0001$, Table S4).” Lines 228-231.

Line 183, Present test statistics for linear model

Pseudo-F scores from the PERMANOVA have now been included alongside the R2. It now reads “Of the listed factors, individuals described the most variation (40%, pseudo-F = 4.59), while country (27%, pseudo-F = 32.76), local environment (7%, pseudo-F = 8.06), sex (4%, pseudo-F = 4.98), diet (4%, pseudo-F = 3.5), and sequencing depth (2%, pseudo-F = 4.13) all described lower amounts.”
Lines 233 – 236.

Line 184: Linear model analyzed effect of country, local environment (urban, suburban, rural – was not discussed/detailed beforehand), sex, diet, and sequencing depth. The authors found that diet explained much less of microbial composition compared to geography, but isn't diet and local environment confounded by geography because each population has singular diets and local environments?

We have listed the local environments in the methods for the dogs from South Africa and India as stated in a previous comment in the methods. The reviewers are correct that there are overlaps here; local environment and diet were subsets within the geographical population. Populations did not necessarily have singular diets (as an example, the dogs from Laos did; the dogs from India did not - this was broken into the shelter population with a known diet and the stray population with an unknown diet). We have addressed this further in a new limitations section to better clarify that these measures are not independent.

There should be more functional figures than just a supplemental table about daidzein metabolism.

There is a figure exploring population similarities in principal coordinate space (Figure S2). We have added additional discussion on the functional results (lactose metabolism) to the supplement.

How did the authors tease apart which taxa were of interest/differentially abundant?

Differentially abundant taxa were identified in comparisons through DESeq2. This has been clarified in the table caption. Taxa of interest were rare taxa that had been previously noted in the literature (i.e., the *Alistipes* genus, which is poorly characterized but may be more common in microbiomes than is currently represented due to biased sampling).

How were the survey data from the SA dogs incorporated into models?

The diet, sex, and local environment (urban, suburban, and farm) were components of the PERMANOVA. Additional tests were run to examine if breed and household produced recognizable differences between dogs, but they did not. This has been added as a point of discussion in the supplement.

Did the authors include individuals as a random effect in the linear models? This is important since some dogs were sampled twice.

The reviewers are correct - we should have included this point initially. This has been added to the first Results section, and reads “Of the listed factors, individuals described the most variation (40%, pseudo-F = 4.59), while country (27%, pseudo-F = 32.76), local environment (7%, pseudo-F = 8.06), sex (4%, pseudo-F = 4.98), diet (4%, pseudo-F = 3.5), and sequencing depth (2%, pseudo-F = 4.13) all described lower amounts.” Lines 233 – 236.

Discussion:

The authors should provide more discussion about the diet experiment and if it follows trends from other canine diet experiments (i.e., Reese et al., 2021 – diet experiment with dogs and wolves).

The authors should also provide more discussion on why India and SA dogs were more similar to US dogs compared to Laos.

We have added a discussion section in the supplement to address the dietary experiment in more detail, as well as the similarities between populations.

Please discuss the limitations with FTA cards as well as limitations in general.

A limitations section has been added to the discussion: Lines 322 – 337.

Lines 254-255, which enzymatic pathways were unique?

An example has now been provided - the line now reads “Outside of homologs unique to various species, few enzymatic pathways were identified that were unique to one population compared to all others, like L-2-oxoglutarate carboxylase in the dogs from Laos.” Lines 311 – 313.

Figures:

Figure 1: It seems reasonable to include Fig. S1 in the main text instead of Fig. 1 (alpha diversity of all populations).

We thank the reviewer for the suggestion and have made this change.

Figure 2: This is a little difficult to look at. Although the authors want to indicate the utility of the Laos data in particular, it might help to make the categories the same in both plots (modern dog, modern human, modern soil, etc.). At the very least, the categories that are shared between panels should use the same color.

Where possible, we've adjusted the figure to include the same nomenclature (not all groups are 1-to-1, and we would like to avoid giving the impression they are). Colors have been redone to match between graphs, as per the reviewer's suggestion.

Figure 3: The authors should include Figure S2 (PCoA of all the populations) in the main text and move this to supplemental. Also, the legend is confusing. Are the coprolites included twice? Once by themselves and then another time "The US dogs include those across all dietary treatments in this study, the coprolite study, and additional individuals across dietary treatments in the Coelho et al. study"? What beta diversity metric is the PCoA based on?

The legend has been clarified to rephrase "the coprolite study" as "the comparative individuals in the Witt et al. study" to indicate the use of these modern dogs in this population, not the coprolites. The caption also explains the usage of the Bray-Curtis beta diversity metric.

Referee: 2

Comments to the Author(s)

The authors have conducted an assessment of gut microbial taxonomy and function for several populations of dogs. They present multiple new datasets, including urban shelter dogs from India, non-pet human-associated dogs from an agricultural community in Laos, pet dogs from South Africa, and beagles that were transitioned onto a diet similar in composition to that of the agricultural dogs. Additionally, they include previously published data from dogs fed high/low protein diets, and ancient DNA from 5 coprolites, where the dogs are presumed to have consumed a mix of wild and domesticated foods. The authors used shotgun metagenomic sequencing followed by taxonomic

profiling, functional profiling, and the identification of microbial “source” populations to assess the relationship modern and ancient dog gut microbiota.

The authors present an outstanding sample set, and I think there is real promise for a substantial contribution to the literature. The increased diversity in the Laotian dogs, is an exciting result, and it’s especially interesting to see that there is a potential overlap between microbial taxa in the coprolites and this non-pet population. However, I have some concerns about your analysis, particularly how you chose to present your results.

1) Generally speaking, I think you need to include more information in your methods and results sections. The text and degree of explanation about how you conducted your analysis is rather compressed. In particular, it would be good if you could provide more detail in your bioinformatics/data analysis section. For example, there are only 3 lines dedicated to the entire functional pipeline. If this is running up against the length limitation, I suggest you include further details in a supplemental methods/results section. If that is not possible, could you provide reproducible code? I checked your github link, but there is no reproducible code there, just a list of tools with descriptions of what they do in a general sense.

We thank the reviewer for their suggestion. We have added a supplemental methods section, adding the requested detail. The bioinformatic pipeline has been discussed in greater depth there. More analyses, initially omitted due to space, were also included in the supplement.

2) It is quite common to find different clustering patterns depending on the normalization/filtration/distance/ordination choices you make. This is of particular concern to me, because your coprolite analysis is heavily dependent on the placement of a single point, coprolite JBGC16, (figures 3, S2A). The ordination in Figure S2A shows that JBGC16 overlaps with samples from India, Laos, and South Africa, whereas the ordination in Figure 3 excludes multiple populations, giving what seems to me an unfair impression of the similarity between the Laotian samples and the coprolites to the exclusion of the non-agrarian populations. If I am misunderstanding something here, please clarify how Figure S2A is consistent with your statement on lines 204-5, “...with the coprolites appearing more similar to the dogs from Laos in coordinate space, regardless of diet.”

We apologize that the figure appears to be misleading. In addressing this and another reviewer’s comments, we have replaced Figure 3 with Figure S2A. The reviewer is correct in noting a level of

nuance that is absent in the sweeping statement quoted. The coprolites overall are more similar to the dogs from Laos - we observe similarities in non-contaminant taxa that are unique to the microbiomes in the dogs from Laos with the coprolites as a whole, like *Bifidobacterium longum* and *Lactobacillus mucosae*. But, the specific coprolite JBGC 16 matches with more populations, likely as a result of its better preservation - we see this play out in matches to taxa like *Tyzzarella nexilis* and *Dorea longicatena*. As the reviewer notes, these taxa are present in the dogs from India, South Africa, and Laos. The sentence has been rewritten as follows to better represent this: “The JBGC16 coprolite clusters independently from other coprolites in PCoA; it contains taxa that match to many of the non-US populations, whereas the remaining coprolites are largely separated and more similar to soil and one dog from Laos (Figure 3).” Lines 255 – 258.

Furthermore, please provide more detail in your methods section about the generation of your beta diversity ordinations. For example, did you filter rare taxa prior to beta diversity analysis? Also, you mention using the DESeq variance stabilizing transformation to normalize your ASV counts. DESeq VST is great for differential abundance testing, but if you look at your transformed ASV counts, I expect that you will see a fair number of negative values. Did you do anything to adjust for this? As I understand it, Bray-Curtis distance is not robust to negative values, so if that’s the case (which it often is) you should use a different distance metric or normalization approach for your ordinations. If you prefer to stick with Bray-Curtis, you should probably use a relative abundance normalization, otherwise weighted unfrac is a better choice for the VST normalization. Take a look at figure 4 in McMurdie and Holmes (2014), which illustrates the relationship between clustering accuracy, effect size and normalization methods (written by the authors of phyloseq, who are the main proponents of using the VST). However you decide to go about this, please justify your choice and provide greater detail in the paper.

We appreciate the reviewer’s attention to detail. Rare taxa were not filtered prior to beta diversity estimates. We recognize that VST can result in negative numbers, but our transformed counts are positive. As recommended, we have provided greater detail for our bioinformatic methods in the aforementioned expanded supplement.

3) I think that your alpha diversity analysis needs further explanation. You describe some populations as “relatively similar.” Are they significantly different? Did you only test the Laotian samples with pairwise tests or did you test all pairs of populations? It would be helpful to include figure S1 along with figure 1 in the main paper to get a sense of the full diversity of these populations prior to lumping

them together. Also, please clearly show on the figures which box plots are significantly different in your paired Wilcoxon tests (or include this information in a supplemental table).

In coordination with other comments, we have swapped Figure 1 and S1 to provide more details in the main text. We conducted pairwise Wilcoxon tests for all combinations, and have provided a table describing these results in the supplement. We have also updated the caption for the new Figure 1, previously Figure S1, to indicate how significant differences change when populations are split.

Secondly, your first mention of a linear model to identify sources of variation across populations is in the results section. You need to describe precisely how this model was built in the methods. Also, how were the amounts of variation calculated? Was there a model testing framework? What software package did you use? etc.

We agree with the reviewer that this language is misleading. We now specify that the sources of variation were tested with a PERMANOVA rather than a linear model, as referenced in the methods. Additional details are provided in the expanded methods in the supplement.

4) You mention finding different taxa in different populations, but I don't see much in the way of quantitative presentation of relative and differential abundances of microbial taxa in your samples. It's hard to tell how meaningful some of your interpretations are without including these results. For example, the caption for table S3 reads "Notable rare taxa or taxa of interest observed in novel study populations. Listed taxa were often differentially abundant in the listed population." Did you find these taxa of interest one time? Or were they common?

We agree with the reviewer about the difficulty in interpretation. We have made an addition to the supplemental table S3 to indicate measures of differential abundance, and the table caption now references the method. The supplemental methods now clarify how differential abundance was identified.

5) In the functional results section, you write, "...the coprolites alone stand out as a single population cluster (Figure S2B)." I don't see how this is the case when the coprolite points in this figure overlap with points from India, Laos, South Africa, and soil.

We agree with the reviewer and have clarified this sentence to reflect these points: ““The JBGC16 coprolite clusters independently from other coprolites in PCoA; it contains taxa that match to many of the non-US populations, whereas the remaining coprolites are largely separated and more similar to soil and one dog from Laos (Figure 3).” Lines 255 – 258.

6) Could you provide further justification/explanation for the use of source tracker? Of course, I understand that you are not making this literal connection, but none of the modern samples are actual sources of the microbes in the coprolites. Perhaps I am wrong here, but I always presumed that source tracker was designed with the expectation of potential transmission between sources and sinks. Again, I might be completely wrong here, so please correct me if am!

SourceTracker is frequently used in ancient microbiome studies (see citations) - as the reviewer notes, in these instances, no literal connection is implied between the sources and sinks. Where they can, these studies (and ours) make use of environmental sources, but often have to rely on less ideal comparative datasets to impute matches. The reviewer is correct in the original intent of SourceTracker, but it has become a useful tool for estimating ancient partitions, though there are limitations, as described in Borry et al., for example.

Appelt S, Armougom F, Le Bailly M, Robert C, Drancourt M. Polyphasic analysis of a middle ages coprolite microbiota, Belgium. *PLoS One*. 2014;9(2):e88376.

Otoni C, Guellil M, Ozga AT, Stone AC, Kersten O, Bramanti B, Porcier S, Van Neer W. Metagenomic analysis of dental calculus in ancient Egyptian baboons. *Scientific Reports*. 2019;9:19637.

Santiago-Rodriguez TM, Fornaciari G, Luciani S, Dowd SE, Toranzos GA, Marota I, et al. Gut Microbiome of an 11th Century A.D. Pre-Columbian Andean Mummy. *PLoS One*. 2015;10(9):e0138135.

Santiago-Rodriguez TM, Narganes-Storde Y, Chanlatte-Baik L, Toranzo GA, Cano RJ. Insights of the dental calculi microbiome of pre-Columbian inhabitants from Puerto Rico. *PeerJ*. 2017;5:e3277.

Tito RY, Knights D, Metcalf J, Obregon-Tito AJ, Cleeland L, Najjar F, et al. Insights from characterizing extinct human gut microbiomes. *PLoS One*. 2012;7(12):e51146.

7) Could you include a supplemental table with details of the different diets? What proportions of these foods are consumed? Can you estimate macronutrient contents?

We thank the reviewer for pointing out this omission. We have included a supplementary table with dietary information and macronutrient contents where available. We also specify that after an initial period of adjustment (no more than a few days), the entirety of the meal was consumed.

McMurdie PJ and Holmes S (2014) “Waste not want not: why rarefying microbiome data is inadmissible. PLOS Computational Biology. 10(4)

We thank the reviewer for recommending this paper. We are aware of the problems it raises in regards to rarefaction and the discussion provided by the manuscript, and have applied its principles to this study.

Appendix B

Referee: 1

Comments to the Author(s).

The authors did a commendable job incorporating previous feedback and/or defending their decisions. There are few revisions we still recommend – the line numbers refer to the track-changes version.

Introduction

- Populations can be identified more than just industrialized vs non industrialized. There are populations that live “in between” (i.e., rural populations near industrialized cities or rapidly developing populations). Indeed, the fact that the authors sample dogs that live in in-between environments (i.e., shelters) highlights this idea and should be another reason to justify using canine microbiomes as a good model for industrialization.

We thank the reviewer for emphasizing this distinction. We initially refer to our geographical populations as “each representing a varying level of industrialization”. We’ve attempted to make this more clear by changing lines 106-7 to read “First, we identify what diversity in the global canine microbiome population looks like, testing the existing hypothesis that increasing degrees of industrialization reduces microbiome diversity, but in dogs.”

- In this vein, there should be a more explicit connection between the ideas of industrialization as a complex system to disentangle microbial contributions and presenting the novel canine microbiome dataset (Lines 22-30). Specifically, the authors may consider justifying why dogs are a good model to disentangle these connections.

We’ve added an additional line (57-9) to make the connection for why canine microbiomes can help disentangle the complexity of this scenario: “A diverse sampling scheme also provides opportunity to explore the complex variation in industrialization around the world, and better identify its influence on canine microbiomes, which are just as exposed to these shifts in diet and environment as their human counterparts.”

- Missing “C” in “Canine” in Line 43

- Line 78, “ther”?

We thank the reviewer for their attention to detail. These were errors in the track changes version only, and are corrected in the main text.

- There should be more discussion on what makes feral and strays different to pets with respect to diet, environment, and/or interactions with humans.

We thank the reviewer for the suggestion. We've updated Table S1 to include this information at a glance, specifying the diet, environment, and human interactions for each population/sub-population, with more detail available in the methods.

Methods

- Lines 163-165, how long did the researchers wait to collect the fecal sample from the 14 dogs brought into the shelter?

We've amended line 125 to read "In addition, researchers collected fecal samples opportunistically from 14 dogs brought into the shelter from the streets as part of routine operations; samples were collected within two to three hours of arrival."

Results

- Line 288, what is "they" referring to – the coprolites or diversity analyses?

To clarify, line 235 now reads "Chao1 alpha diversity indicates that the coprolites vary greatly in alpha diversity, though the coprolites also include the samples with the lowest measures (Figure 1)."

- While the authors have presented more detail in their metagenomics methodology, there should be more space dedicated to metagenomics results – for example, did the PERMANOVA also explain variation in the functional potential of the canine microbiomes?

Additional metagenomic information on the populations less-discussed in the main text can be found in the supplement. We have added the following section to the supplement to address the PERMANOVA based on functional data "A PERMANOVA to explore sources of variation in the canine microbiomes based on their functional profiles was generally similar to the taxonomic breakdown. Individuals described the largest component of variation (53%, pseudo-F = 3.60), followed by country (25%, pseudo-F = 39.04), local environment (10%, pseudo-F = 9.93), sex (6%, pseudo-F = 8.79), diet (1%, pseudo-F = 11.45), and sequencing depth (1%, pseudo-F = 9.76)."

- What about diet in other populations apart from beagles?

Outside of the beagles, only the dogs from South Africa and India (shelter) had known diets. While minimal results were determined from the recorded diets of the dogs from South Africa, there was an interesting point on lactose metabolism found in both populations, which is described in the supplement. While interesting, this point also lacks as much supporting evidence as the point made in the main text on daidzein metabolism, hence the decision to relegate one part to the supplement.

Discussion

- Much of the discussion is on the data from the Laos dogs and the coprolites – there should be more discussion on the microbiomes from other dog populations

Due to space constraints, it is not possible to fully discuss each population within the main text. However, in accordance with comments from another reviewer, we have shifted the balance of the text to attempt to strike a compromise.

- FTA cards preserve samples effectively, but the resulting communities differ from ethanol-preserved samples (line 299). This sentence could just use some clarification on this point (and a reference).

We agree with the reviewer; comparable was meant to refer to the quality of resulting data, rather than directly comparing results between methods. We have added a reference to lines 309-11 and reworded the statement to be more explicit; it now reads “FTA cards have been used for fecal samples and produce consistent results, but they are not the gold standard of fresh-frozen samples, and will produce biases in microbiome reconstruction; however, the sample collection for this study is uniform (86).”

Figures

- Figure 3, where are the samples from SA and India in the PCoA plot?

We apologize; this was an error in labeling figures and submission, and should instead be a PCoA of all the populations as recommended by both reviewers in the previous round of comments. This has been corrected.

Referee: 2

Comments to the Author(s).

The authors have substantially improved the paper from the first version that I read. In particular, I appreciate that they have included the differential abundance table and the expanded methods in the

supplement. However, I still have concerns regarding their over-interpretation of the ordinations and justification for similarity between the coprolites and samples from Laos. I have responded to your responses to my queries in the corresponding numerical order.

1) The additional methodological details that you have provided in the expanded supplement is much improved and mostly satisfies my request.

2A) I am still having trouble with how you justify your interpretation that. “[t]he coprolites overall are more similar to the dogs from Laos.” Yes, the point you make here in your response that coprolite JBGC16 shares some taxa with samples from Laos is reasonable. However, I do not see any statistical evidence showing that the abundances of these taxa are similar in the two groups. Do you mean that they are simply present at least once in JBGC6 and at least one Laotian sample? Second, I do not agree with your use of the PCoAs (any of them) to demonstrate similarity between the ancient dogs and the dogs from Laos. Ultimately, you are relying on one sample to argue this point. I appreciate that you have softened the language in the text, but in the caption of figure S3A, you still write that, “dogs from Laos show more similarity to coprolites, with JBGC16 indicated by the red arrow.” If your argument is that the position of JBGC16 in the PCoA demonstrates that the coprolites are more similar to the samples from Laos, one could also use the single sample from Laos that clusters with the US samples to declare that all the dogs from Laos are similar to the US dogs (which is of course, false). The coprolites might well be more similar to the samples from Laos—your source tracker results show some evidence this—but your PCoAs do not. What I am saying here is that you cannot use the placement of a single coprolite in the PCoA while ignoring the placement of the rest of the coprolites to justify the similarity of the coprolites as a group to the samples from Laos as a group (especially given their positioning in Figure S2A on Axis 1, which explains over twice the variance of axis 2). Also, Figure 3 appears identical to figure S2B minus the arrow. You should replace figure 3 with Figure S2A, which shows the more fine scale differences, or include both in the manuscript.

We agree that our focus on a single coprolite has led to some overstatement in the caption, and have revised the wording to avoid over-extrapolating from the PCoAs. The quoted text has been replaced with “The dogs from Laos demonstrate the greatest diversity, with the US dogs maintaining a small cluster despite shifts in diet. The red arrow indicates JBGC 16, the best preserved coprolite.”

2B) It is encouraging that the VST transformed Bary-Curtis values are positive. However, my criticism of the use of DEseq VST with Bray-Curtis distances was two pronged, and the authors have not addressed my second point, illustrated in Figure 4 of McMurdie and Holmes (2014): in absence of a large effect size, Bray-Curtis distance is not accurate when combined with DEseq variance stabilization. Again, if you want to use the VST, you should be generating weighted unifrac distances,

not Bray-Curtis distances, unless you have demonstrated a large effect size. Normally, I would not be insistent about this issue—by and large ordinations look the quite similar with different distances/transformations—but your interpretation of the similarity between the coprolites and the samples from Laos is dependent on the position of the single point, JBGC16, and as such, I don't think it is appropriate to rely on the accuracy of its positioning in the ordination when the combination of data transformation and distance metric used to generate that point has been demonstrated not to be very accurate. At the very least, I would like to see a version of figure 3 that uses weighted unfrac distance and VST or Bray-Curtis distance with relative abundance transformation.

We appreciate the reviewer's insistence on the need for increased accuracy in statistical methodology; we are in complete agreement on the improvements that could be offered by the usage of the UniFrac distance metric, versus the Bray-Curtis metric used here. Unfortunately, there are non-trivial barriers to implementing this method with our shotgun metagenomic data. Shotgun metagenomic analysis tools are still developing; most often, development of these tools focuses on human microbiomes, as these provide the most robust datasets for testing. This creates biases and complications when attempting to translate analyses to novel, largely undescribed populations. We have unsuccessfully attempted to utilize many such tools, before selecting the methodology described in the paper, based on our needs for a comparative analysis with degraded, ancient DNA samples. Without restricting the quantity of sequences used for comparative purposes to a common marker sequence, we do not currently have an informative means to calculate distances with the UniFrac metric, which in turn eliminates most of the aDNA sequence data available from our samples.

We recognize the limitations this produces in the results; we have referenced this in our limitations paragraph (lines 312-3), and noted it quantitatively in Table S1. We also recognize the difficulty in demonstrating a large effect size in a multi-factorial microbial study. To accommodate these concerns, we have revised the discussion to remove reference to PCoA based similarities between populations. We continue to include the PCoAs in the interest of demonstrating visually the varying breadth of diversity in populations.

The Figure 3 used in the main text was an error on our end; based on previous comments from reviewers, it was meant to be the full PCoA (for all samples). We apologize for the misunderstanding and have corrected this error.

3) I appreciate the inclusion of the additional tests, clarification of the alpha diversity analysis, and PERMANOVA. Much improved.

4) Regarding the quantitative analysis of differential abundance, I presume you mean table S5, and not S3 (or did I mix those up in my initial response? Regardless, much better, thanks. Did you also conduct this analysis on the coprolites? I don't see any reference to them. If not, how are you determining the similarity between the coprolite taxa and other populations addressed in point #2A? For example, in the manuscript, you write, "[t]he coprolites and dogs from Laos are both marked by an increased abundance of Enterococcus and Lactococcus compared to other canine microbiomes." I don't see how is this demonstrated statistically.

We thank the reviewer for spotting this omission; the coprolite data was part of this analysis and is now included in the table.

5) The revised text is better, but the PCoA issues discussed elsewhere need to be addressed.

6) Thank you for this explanation and the list of papers. It is very helpful.

7) Thank you for the additional diet table. Regarding the comment that you have applied the principles of McMurdie and Holmes (2014) to this paper, I don't see how that is the case, based on my comment in point #2B.

Appendix C

Board Member: 1

Comments to Author:

Many thanks to the authors for submitted a revised version of their manuscript. Each time I read over this study, my enthusiasm for this interesting and timely research resurfaces. I anticipate this study will make a valuable contribution to the field.

The reviewers and myself find the revised version of the manuscript to be much improved. With one minor exception, the revisions carried out were satisfactory. However, I draw attention to a point raised by a reviewer regarding a PCoA based on Bray-Curtis and relative abundances. Please provide this in the final version ahead of publication, or provide a detailed explanation as to why this isn't possible. And as a very minor point, I also suggest the authors use a lower case "v" in the heading: Diet versus Environment.

We thank the board member for their enthusiasm over the submitted work. We apologize for the oversight regarding the reviewer's unaddressed comment, and have included the requested figure in the supplement, and make reference to it in the main text. We have also made the typographical edit suggested.

Reviewer(s)' Comments to Author:

Referee: 2

Comments to the Author(s)

Thank you for the adjustments that you have made to the manuscript. I think it has been improved. Regarding point 2B, I wrote, that "At the very least, I would like to see a version of figure 3 that uses weighted unifrac distance and VST or Bray-Curtis distance with relative abundance transformation." Thank you for explaining your case-specific issue with unifrac distances. I see your point. However, I fail to see why you disregarded my alternative recommendation that you show me a PCoA based on Bray-Curtis and relative abundances, which is certainly possible to run. Nonetheless, I will not press the point further, since you have toned down the interpretation of the PCoA, and overall it's a great paper.

We thank the reviewer for their multiple rounds of feedback; we agree the manuscript has been improved, in large part thanks to this effort. We apologize for missing the alternative recommendation; this was not intentional, and have included the requested graph in the supplement, with references to it in the main text.